# CLDyB: Towards Dynamic Benchmarking for Continual Learning with Pre-trained Models

**Shengzhuang Chen**[1], **Yikai Liao**[2], **Xiaoxiao Sun**[2,3], **Kede Ma**[1,*], **Ying Wei**[4,*]
[1]City University of Hong Kong    [2]Nanyang Technological University
[3]Australian National University    [4]Zhejiang University
`shengzhuang.chen@my.cityu.edu.hk`
`kede.ma@cityu.edu.hk`
`ying.wei@zju.edu.cn`

## Abstract

The advent of the foundation model era has sparked significant research interest in leveraging pre-trained representations for continual learning (CL), yielding a series of top-performing CL methods on standard evaluation benchmarks. Nonetheless, there are growing concerns regarding potential data contamination during the pre-training stage. Furthermore, standard evaluation benchmarks, which are typically static, fail to capture the complexities of real-world CL scenarios, resulting in saturated performance. To address these issues, we describe CL on dynamic benchmarks (CLDyB), a general computational framework based on Markov decision processes for evaluating CL methods reliably. CLDyB dynamically identifies inherently difficult and algorithm-dependent tasks for the given CL methods, and determines challenging task orders using Monte Carlo tree search. Leveraging CLDyB, we first conduct a joint evaluation of multiple state-of-the-art CL methods, leading to a set of commonly challenging and generalizable task sequences where existing CL methods tend to perform poorly. We then conduct separate evaluations of individual CL methods using CLDyB, discovering their respective strengths and weaknesses. The source code and generated task sequences are publicly accessible at `https://github.com/szc12153/CLDyB`.

## 1 Introduction

The field of machine learning is undergoing a paradigm shift driven by the emergence of foundation models—large-scale neural networks pre-trained on massive datasets—that demonstrate remarkable adaptability across a wide range of downstream tasks. Within this transformative landscape, continual learning (CL), a computational methodology for incrementally updating models while retaining prior knowledge (Ring, 1997), faces both unprecedented opportunities and critical challenges. Opportunities arise from the synergistic relationship between CL and foundation models. On one hand, foundation models stand to gain from CL techniques by continually adapting to evolving data distributions without catastrophic forgetting. On the other hand, CL methods can leverage the general-purpose and robust representations embedded within foundation models as powerful initializations, potentially accelerating convergence and improving performance on sequential tasks. Challenges, however, primarily stem from the inadequacy of conventional CL benchmarks (Rebuffi et al., 2016; Hendrycks et al., 2020) in providing reliable evaluations of CL methods with foundation models due to two critical limitations.

The first is **data contamination**. The exponential growth in the volume of pre-training data for foundational models elevates the risk of prior exposure to downstream CL tasks. This has precipitated performance saturation in current benchmarks, diminishing their utility of comparing CL methods through marginal performance gains. More fundamentally, this overlap in data distributions calls into question whether recent progress in CL stems from genuine algorithmic developments or merely reflects the exploitation of more advanced foundation models (Janson et al., 2022; Galashov et al., 2023). The second is **performance saturation**. Current benchmarks are inherently static,

---

*Corresponding authors.

which assume that tasks are randomly sampled from fixed datasets and presented in an unstructured, sequential order. This is a clear oversimplification of real-world CL scenarios, which involve dynamic and evolving task sequences that are significantly more varied and challenging. Consequently, CL methods may overfit these simplified benchmarks while failing to generalize in more complex and practically demanding environments.

In light of the pressing demand for reliable evaluation protocols, we introduce CL on dynamic benchmarks (CLDyB), a computational benchmarking framework designed to accelerate algorithmic progress in CL with foundation models. Specifically, we formulate dynamic benchmarking for CL as an infinite-state Markov decision process (MDP, Puterman, 1994), where the sequential task selection and model adaptation are optimized over a potentially unbounded time horizon. The state at time step $t$ encapsulates the consolidated model parameters after incremental training on all observed tasks up to $t$. This parameter vector serves as a compact representation of the model's current knowledge retention (*i.e.*, stability) and its adaptability to new tasks (*i.e.*, plasticity). The action corresponds to selecting a task from a combinatorially large pool of candidate tasks, followed by stochastic optimization of model parameters on the selection task. The reward is defined as the immediate, quantitative measure of how challenging the selected task is given the current state.

Due to the infinite state space, the vast action space, and the high computational cost of reward evaluation, finding an exact solution to this MDP (for instance, via dynamic programming) is computationally intractable. To address this, the proposed CLDyB first reduces the action space using greedy task sampling and clustering. This yields a much smaller feasible set of diverse and difficult tasks that are more likely to challenge the original foundation model (as a way of mitigating data contamination) and the continually updated models at each time step. Second, it employs Monte Carlo tree search (MCTS, Coulom, 2006) with a truncated lookahead to identify the (sub)optimal task at each time step.

We instantiate CLDyB for class-incremental learning in visual recognition (Zhou et al., 2024b)—one of the most prominent challenges in CL research. We first apply CLDyB jointly to a group of representative CL methods, generating a standardized set of challenging and generalizable task sequences that yield reliable and indicative evaluation results. We then perform multi-dimensional assessment of individual CL methods using CLDyB, assessing critical dimensions such as classification accuracy, robustness to task sequence variations, and memory efficiency. This allows us to inspect distinct performance characteristics and vulnerabilities of different CL methods.

In summary, our principal contributions include

- **CLDyB**: A general computational benchmarking framework designed to inform reliable progress in CL, especially with pre-trained models;

- **Standardized CL benchmark**: CLDyB establishes a set of challenging and generalizable task sequences for both current and future CL methods;

- **Multi-dimensional assessment**: CLDyB enables comprehensive assessment of individual CL methods across multiple dimensions.

## 2 PRELIMINARIES

Class-incremental learning aims to build a generic classifier capable of recognizing an expanding set of classes by incrementally integrating new knowledge while maintaining performance on previously learned tasks (Zhou et al., 2024b). More formally, a CL algorithm $\text{CL}(\cdot)$ incrementally trains a parameterized classifier $f$ on a sequence of $N$ classification tasks, collectively denoted as $\mathcal{T}_{<N+1} := \{\mathcal{T}_1, \mathcal{T}_2, \ldots, \mathcal{T}_N\}$. Each task $\mathcal{T}_t$ contains $|\mathcal{T}_t|$ (image, label) pairs $\{\boldsymbol{x}_t^{(j)}, y_t^{(j)}\}_{j=1}^{|\mathcal{T}_t|}$, divided into training, validation, and test subsets. Note that all tasks have disjoint sets of class labels. At time step $t$, the classifier is updated according to $f_t = \text{CL}(f_{t-1}, \mathcal{T}_t)$, where $f_0$ denotes the pretrained foundation model. A fundamental constraint in CL is that data from past or future tasks remain inaccessible while training on the current task. This presents a central challenge: Training $f$ to incrementally recognize new classes (*i.e.*, high plasticity), while mitigating the catastrophic forgetting of previously learned classes (*i.e.*, high stability). To quantify the plasticity and forgetting of $f_t$, we employ two standard metrics: Average learning accuracy (ALA, Riemer et al. 2018) and

the average forgetting measure (AFM, Chaudhry et al., 2018):

$$\text{ALA}\left(f_t, \mathcal{T}_{<t+1}\right) = \frac{1}{t} \sum_{k=1}^{t} \text{Acc}\left(f_k, \mathcal{T}_k\right), \tag{1}$$

$$\text{AFM}\left(f_t, \mathcal{T}_{<t+1}\right) = \frac{1}{t-1} \sum_{k=1}^{t-1} \text{Acc}\left(f_k, \mathcal{T}_k\right) - \text{Acc}\left(f_t, \mathcal{T}_k\right). \tag{2}$$

$\text{Acc}(f, \mathcal{T})$ represents the empirical classification accuracy of classifier $f$ on the test split of task $\mathcal{T}$. Assuming there are $M$ CL methods, each associated with its respective parameterized classifier at time step $t$, denoted as $\{\text{CL}^{(i)}\}_{i=1}^{M}$ and $\mathcal{F}_t := \{f_t^{(i)}\}_{i=1}^{M}$, we compute the ALA and AFM scores by further averaging over these $M$ models, *i.e.*, $\text{ALA}(\mathcal{F}_t, \mathcal{T}_{<t+1}) = \frac{1}{M} \sum_{i=1}^{M} \text{ALA}(f_t^{(i)}, \mathcal{T}_{<t+1})$ and $\text{AFM}(\mathcal{F}_t, \mathcal{T}_{<t+1}) = \frac{1}{M} \sum_{i=1}^{M} \text{AFM}(f_t^{(i)}, \mathcal{T}_{<t+1})$.

## 3 PROPOSED FRAMEWORK: CLDYB

In this section, we first give the general problem formulation of dynamic benchmarking for CL, and then introduce the proposed CLDyB as a specific instance.

### 3.1 PROBLEM FORMULATION

Our core idea is to *sequentially construct algorithm-dependent task sequences that consistently challenge current CL methods*. Mathematically, this sequential task construction can be formulated as an infinite-state MDP $(\mathcal{S}, \mathcal{A}, R)$, with a potentially unbounded time horizon (*i.e.*, the task length is unknown or $N \to \infty$). At time step $t$, the observable state comprises the set of $M$ CL models $\mathcal{F}_{t-1} \in \mathcal{S}$, each having been sequentially trained on a common sequence of selected tasks up to time step $t$. We assume a policy $\mu_t : \mathcal{S} \mapsto \mathcal{A}$ that maps the current state to an action, which involves selecting a specific task $\mathcal{T}_t$ from a feasible set of tasks (*i.e.*, the action space $\mathcal{A}_t$). After that, a probabilistic state transition is induced from $\mathcal{F}_{t-1}$ to $\mathcal{F}_t$ through the stochastic optimization of $\mathcal{F}_{t-1}$ using their respective CL methods on the selected task: $f_t^{(i)} = \text{CL}^{(i)}(f_{t-1}^{(i)}, \mathcal{T}_t)$, for $i \in \{1, 2, \ldots, M\}$. This transition incurs an immediate reward:

$$R\left(\mathcal{F}_t, \mathcal{T}_{<t+1}\right) = \text{AFM}\left(\mathcal{F}_t, \mathcal{T}_{<t+1}\right) - \text{ALA}\left(\mathcal{F}_t, \mathcal{T}_{<t+1}\right), \tag{3}$$

which encourages forgetting while inhibiting plasticity, thereby facilitating the identification of challenging tasks to mitigate data contamination (Yeom et al., 2017). Consequently, our objective translates to finding the optimal policies $\pi^\star = \{\mu_t^\star(\mathcal{F}_{t-1})\}_{t=1}^{N}$, that maximize the cumulative rewards:

$$\pi^\star = \arg\max_{\pi} \underbrace{\lim_{N \to \infty} \sum_{t=1}^{N} \alpha^{t-1} R(\mathcal{F}_{t-1}, \mu_t(\mathcal{F}_{t-1}))}_{J_\pi(\mathcal{F}_0)}, \tag{4}$$

where $\alpha \in (0, 1]$ is a discount factor, $\mathcal{F}_0 = \{f_0^{(i)}\}_{i=1}^{M}$, and $J_\pi(\mathcal{F}_0)$ is the initial state value function.

Solving Problem (4) to obtain an exact solution presents two primary computational challenges. First, the solver shall efficiently navigate the vast and discrete set of feasible tasks (*i.e.*, $\mathcal{A}_t$) at each time step, which grows combinatorially with problem complexity (*e.g.*, the total number of classes). Second, the solver shall accurately and efficiently evaluate the value function (*i.e.*, $J_\pi(\mathcal{F}_0)$) to facilitate challenging task identification. Nevertheless, the complexity of this evaluation can vary significantly based on the chosen continual learning (CL) method and task properties, such as the number of classes per task. Moreover, the computation may become intractable due to the infinite state space and potentially unbounded time horizon. In the subsequent subsections, we describe our proposed solution, CLDyB, which effectively addresses these challenges through two key strategies: 1) Greedy task sampling and clustering to construct a much smaller candidate task set, and 2) MCTS to identify the (sub)optimal task. Fig. 1 illustrates the system diagram of CLDyB.

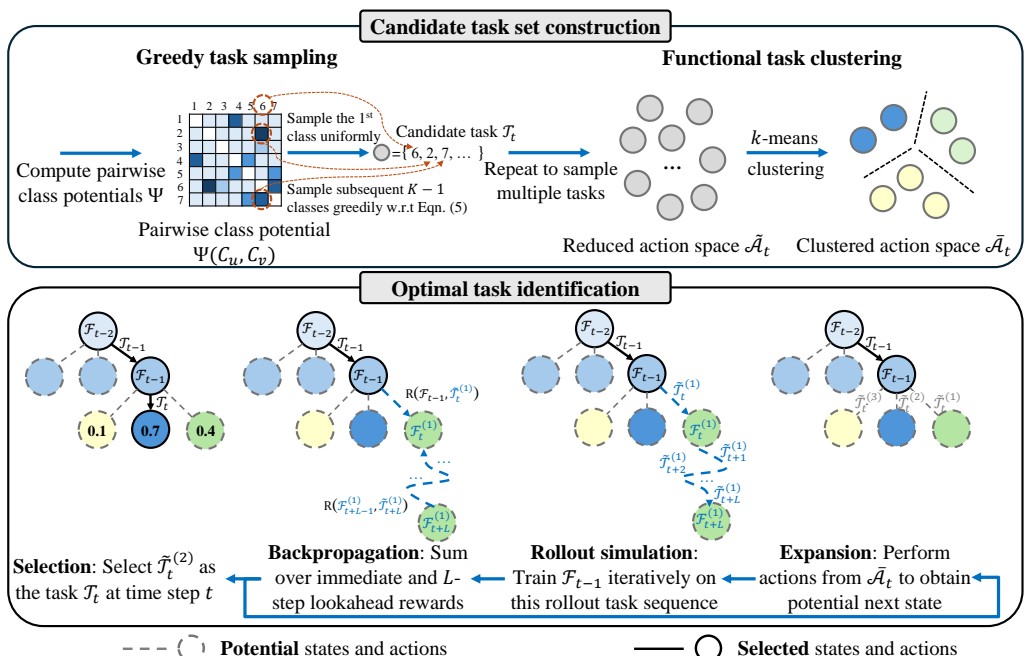

Figure 1: System diagram of the proposed CLDyB for dynamically constructing task sequences that challenge current CL methods. At time step $t$, CLDyB first performs **candidate task set construction** by greedy task sampling (see Eqn. (5)) and functional task clustering. This results in a reduced and clustered action space $\bar{\mathcal{A}}_t$, facilitating task evaluation. It then performs **optimal task identification** by maximizing the estimated current state value function (see Eqn. (7)) using MCTS, which consists of four steps: 1) Expansion, 2) rollout simulation, 3) backpropagation, and 4) selection. The pseudocode for CLDyB can be found in Algorithm 3 of the Appendix.

## 3.2 CANDIDATE TASK SET CONSTRUCTION

Given a data pool $\mathcal{D}_t$ of $C_t$ classes for class-incremental learning at time step $t$, we define a task $\tilde{\mathcal{T}}_t$ as a subset of images containing $K$ classes, where $K < C_t$. Consequently, the number of possible tasks in $\mathcal{A}_t$ is given by the binomial coefficient $\binom{C_t}{K}$, which grows rapidly as $C_t$ increases. This makes an exhaustive search over $\mathcal{A}_t$ infeasible, as each task evaluation often necessitates a full cycle of continual model training, significantly increasing the overall computational cost.

**Greedy task sampling.** To construct a more compact candidate task set $\tilde{\mathcal{A}}_t$, we propose a greedy task sampling strategy that leverages the intuition that task difficulty in multi-class classification is inversely correlated to the separability of class pairs in feature space (He et al., 2020). Specifically, we define the sampling probability over a task $\tilde{\mathcal{T}}_t$ to be proportional to the product of its pairwise class potential functions:

$$p(\tilde{\mathcal{T}}_t) \propto \prod_{u,v \in \{1,\dots,K\}} \Psi(\mathcal{C}_u, \mathcal{C}_v), \text{ where } \Psi(\mathcal{C}_u, \mathcal{C}_v) = \frac{1}{M} \sum_{i=1}^{M} s\left(\text{CosSim}\left(\boldsymbol{u}_t^{(i)}, \boldsymbol{v}_t^{(i)}\right)\right). \quad (5)$$

Here, $\mathcal{C}_u$ and $\mathcal{C}_v$ represent the subsets of images belonging to classes $u$ and $v$, respectively. The prototypes $\boldsymbol{u}_t^{(i)}$ and $\boldsymbol{v}_t^{(i)}$ are computed by the mean feature representations of these subsets using $f_{t-1}^{(i)}$. CosSim$(\cdot, \cdot)$ computes the cosine similarity between two feature representations, and $s(\cdot)$ is a linear rescaling function that maps the $M$ similarity scores to the range $[0, 1]$, where the maximum value is set to 1 and the minimum to 0. This additional rescaling ensures that the computed pairwise potentials are well-suited for probability computation (*i.e.*, preventing negative values). By assigning higher sampling probabilities to tasks with larger products of pairwise potentials, we prioritize challenging tasks, for which the classifiers $\mathcal{F}_{t-1}$ struggle to discriminate between the included classes in their respective feature spaces. This will force $\mathcal{F}_{t-1}$ to focus on refining their feature

representations for more difficult cases, thereby improving their capacity to handle more complex class relationships.

Drawing inspiration from (Kohli et al., 2013), we introduce a greedy sampler that sequentially selects $K$ classes to form a task $\tilde{\mathcal{T}}_t$. Initially, to ensure task diversity, the first class is uniformly sampled from all feasible classes in the data pool $\mathcal{D}_t$. Subsequently, during the $k$-th iteration, the class $u_k^\star$ is chosen to maximize the product of pairwise potentials with respect to the previously selected $k-1$ classes in $\mathcal{V}_{k-1}^\star$:

$$u_k^\star = \arg\max_u \prod_{v \in \mathcal{V}_{k-1}^\star} \Psi(\mathcal{C}_u, \mathcal{C}_v), \text{ for } k \in \{2, \ldots, K\}, \tag{6}$$

and we incorporate $u_k^\star$ into $\mathcal{V}_{k-1}^\star$ to form $\mathcal{V}_k^\star$. This greedy sampling iterates until $K$ distinct classes are identified. The entire procedure is further repeated multiple times to generate the candidate task set $\tilde{\mathcal{A}}_t$. A detailed description of our sampler is provided in Algorithm 1.

**Functional task clustering.** The sampled tasks in $\tilde{\mathcal{A}}_t$ are likely to exhibit a skewed distribution with respect to their enabled functional skills of classifiers. This is because the $M$ classifiers naturally form distinct clusters based on their functional skills. Consequently, uniform task sampling from $\tilde{\mathcal{A}}_t$ disproportionately encourages selecting tasks from the cluster characterized by the dominant functional skills, thereby limiting exposure to tasks that would challenge classifiers in diverse ways. To enhance task diversity in functional space and promote more efficient task evaluation, we propose additionally clustering tasks in $\tilde{\mathcal{A}}_t$ based on their induced functional skills, followed by ancestral sampling to populate a further condensed candidate task set $\bar{\mathcal{A}}_t$.

Specifically, for each task, we compute an $M$-dimensional feature vector by evaluating the average negative log-likelihood of predictions made by $M$ $k$-nearest neighbors ($k$-NN) classifiers with feature representations from $\mathcal{F}_{t-1}$. We partition tasks into functionally coherent clusters using the $k$-means algorithm, operating on the derived task representations. The final condensed candidate task set $\bar{\mathcal{A}}_t$ is populated through ancestral sampling: First uniformly selecting a cluster and then randomly sampling a task from the chosen cluster. A detailed description is provided in Algorithm 2.

### 3.3 OPTIMAL TASK IDENTIFICATION

Despite substantial action space reduction via greedy task sampling and clustering, optimizing the initial state value function defined in Eqn. (4) remains computationally challenging. This difficulty arises from the combinatorial complexity of task ordering and the long-term interdependencies between task selection and model adaptation. To address these, we employ MCTS (Coulom, 2006), a widely adopted approach for online sequential decision-making (Puterman, 1994), particularly well-suited to our CL setup. Specifically, MCTS approximates the current state value $J_\pi(\mathcal{F}_t)$ within a search tree, comprising four steps. 1) Rollout simulation: Recursive tree growing via MC simulations until a predefined computational budget is exhausted; 2) Backpropagation: Update values and visit counts for all nodes in the visited path; 3) Selection: Choose the task $\mathcal{T}_t$ that leads to the state with the maximum estimated immediate value; 4) Expansion: Train all classifiers continually on the selected task $\mathcal{T}_t$ and transit from the current state $\mathcal{F}_{t-1}$ to the next state $\mathcal{F}_t$.

Unfortunately, in open-ended CL environments, MC simulations are resource-intensive due to 1) the necessity of $(N-t)$-step rollouts to evaluate the value function $J_\pi(\mathcal{F}_t)$, involving continual model training across tasks at all these steps, and 2) the indefinite task length $N$. We propose approximating the value function using a truncated horizon approach, leading to

$$J_\pi(\texttt{CL}(\mathcal{F}_{t-1}, \mathcal{T}_t)) \approx \max_{\tilde{\mathcal{T}}_t \in \bar{\mathcal{A}}_t} R(\mathcal{F}_{t-1}, \tilde{\mathcal{T}}_t) + \sum_{k=0}^{L-1} R(\texttt{CL}(\mathcal{F}_{t-1+k}, \tilde{\mathcal{T}}_{t+k}), \tilde{\mathcal{T}}_{t+k+1}), \tag{7}$$

where $\mu_t = \mathcal{T}_t \in \pi$ represents the optimized policy at time step $t$. The terms $\tilde{\mathcal{T}}_{t+k}$, for $k \in \{1, \ldots, L\}$ are sampled uniformly from their respective feasible action spaces, and are excluded from the set of optimization variables (i.e., they are not included in the $\arg\max$ operation). The hyperparameter $L$ controls the trade-off between computational tractability and approximation fidelity. For simplicity, we set $L = 1$ and defer the adoption of advanced value function approximation methods, such as value network learning (Silver et al., 2016), to future research.

# 4 RELATED WORK

**Class-incremental learning.** Recent advances in CL with foundation models reveal two key advantages: inherent resistance to catastrophic forgetting and strong generalization to downstream tasks (Ostapenko et al., 2022; Zhang et al., 2023), rendering it a rapidly evolving research frontier. A prominent technical direction is to integrate CL methods with parameter-efficient fine-tuning strategies, yielding representative methods like orthogonal projection (Liang & Li, 2024; Qiao et al., 2024), model expansion (Wang et al., 2022; Smith et al., 2022; Wang et al., 2023a), and model ensembling (Gao et al., 2023; Zhou et al., 2024a). Alternatively, representation-based methods focus on maintaining stable feature representations within foundation models. This is often achieved by freezing the backbone after the initial task learning (Zhou et al., 2023a) or employing low learning rates (Zhang et al., 2023). Non-parametric classifiers are then progressively constructed, which can be enhanced by random projections (McDonnell et al., 2023) or the inclusion of intermediate representations (Ahrens et al., 2023). A more detailed discussion of conventional CL methods is deferred to Appendix B.1.

**Dynamic benchmarking.** The rapid advancement of foundation models, particularly large language models (LLMs), has necessitated a re-evaluation of traditional static benchmarks, often inadequate in providing reliable assessment due to issues like data contamination (Shi et al., 2023; Zhou et al., 2023b) and oversimplification of real-world testing scenarios (Kiela et al., 2021; McIntosh et al., 2024). Dynabench (Kiela et al., 2021) and DynaBoard (Ma et al., 2021) propose to continuously evolve test sets with crowd-sourced human labels to enable more accurate LLM evaluations. Recognizing the substantial costs involved in human labeling, recent efforts have been made to automate the process of dynamic test set generation, leveraging directed acyclic graphs (Zhu et al., 2023) and multiple agents (Wang et al., 2024). Additionally, dynamic programming has been used for test set selection, enhancing the scalability of evaluating a large number of models at ten thousand scales (Prabhu et al., 2024). The proposed CLDyB represents one of the initial endeavors to establish dynamic benchmarking protocols for CL methods, particularly those with foundation models.

# 5 EXPERIMENTS

In this section, we first present the experimental setups, and then evaluate nine state-of-the-art CL methods both jointly and separately using CLDyB, followed by insightful analysis.

## 5.1 EXPERIMENTAL SETUPS

**Data pool.** To build the playground for CLDyB, we assemble a total of 26 datasets, consisting of $2,505,185$ images across $2,403$ categories. These datasets are categorized into two main groups: photographic image datasets for the main experiments and AI-generated image datasets for additional analysis. The included classes cover a broad spectrum of domains, levels of granularity, cultural contexts, and time periods. Details are provided in Appendix A.

**CL methods.** We select a total of nine CL methods for evaluation based on two main criteria: 1) Competitive performance that prioritizes recently published methods, achieving state-of-the-art results on static CL benchmarks, and 2) sufficient representativeness that ensures the selected methods encompass a wide range of design philosophies.

**Implementation details.** All experiments adhere to the standard class-incremental learning protocols, with results evaluated using three widely adopted metrics: 1) Average accuracy (Acc), 2) average retention[1] (AR, Chaudhry et al. 2018), and 3) ALA (Riemer et al., 2018). Higher values indicate better CL performance but also suggest that the benchmark presents a lower level of difficulty for CL methods being evaluated. During training, hyper-parameters for each method are selected using the validation sets of the first three tasks, following Chaudhry et al. (2018). Each task in the CLDyB sequences is a 20-category classification problem (*i.e.*, $K = 20$) with a sequence length of ten (*i.e.*, $N = 10$). All CL methods employ ViT-Base-Sup21K (Dosovitskiy et al., 2021) as the foundation model. All main experiments are repeated across three random runs, and mean results are reported. To ensure a fair comparison of task selection strategies, the first task is randomly chosen

---

[1]Equivalent to the negation of AFM in Eqn. (2).

Table 1: Final average accuracy (%) of CL methods on various benchmarks. "‡" indicates results directly copied from the respective papers, while "∗" highlights statistically significant results. AFEC and CLSER are excluded from comparison here due to differences in backbone architectures.

| Method | CIFAR-100[‡] | ImageNet-R[‡] | CLDyB (Ours) | Held-out |
|---|---|---|---|---|
| ER (Rolnick et al., 2018) | 67.9 | 55.1 | 54.8 | 79.9 |
| DualPrompt (Wang et al., 2022) | 86.5 | 68.1 | 41.9 | 70.8 |
| LAE (Gao et al., 2023) | 85.6 | 72.7 | 48.1 | 71.1 |
| HiDe-Prompt (Wang et al., 2023a) | **92.6** | 75.1 | **62.5** | **84.9** |
| SLCA (Zhang et al., 2023) | 91.5 | 77.0 | 56.2 | 80.3 |
| RanPAC (McDonnell et al., 2023) | 92.2 | **78.1** | 56.9 | 81.0 |
| PGP (Qiao et al., 2024) | 86.5 | 69.3 | 44.0 | 68.7 |
| SRCC (per column & held-out) | 0.643 | 0.643 | **0.964**∗ | N/A |
| KRCC (per column & held-out) | 0.429 | 0.429 | **0.905**∗ | N/A |

and fixed, maintaining consistent initial conditions. Additional details on the selected CL methods can be found in Appendix B.2.

## 5.2 MAIN RESULTS

**CLDyB identifies commonly challenging task sequences.** We first conduct a joint evaluation of five CL methods, *i.e.*, AFEC (Wang et al., 2021), CLSER (Arani et al., 2022), HiDe-Prompt (Wang et al., 2023a), RanPAC (McDonnell et al., 2023), and PGP (Qiao et al., 2024) using CLDyB, while reserving the remaining four CL methods, *i.e.*, ER (Rolnick et al., 2018), DualPrompt (Wang et al., 2022), LAE (Gao et al., 2023), and SLCA (Zhang et al., 2023), for independent testing. The results, presented in Figs. 2 and 8, demonstrate that the CLDyB sequences pose significant challenges not only for the five CL methods subjected to CLDyB but also for the four CL methods that were not exposed to it. Notably, compared to random task sequences drawn from the data pool, performance on the CLDyB sequences results in a 26% reduction in Final Acc and a 9% reduction in Final AR across the evaluated methods, as shown in Fig 6. These cross-validation results underscore the generalizability of the CLDyB sequences, manifesting themselves as a strong CL benchmark.

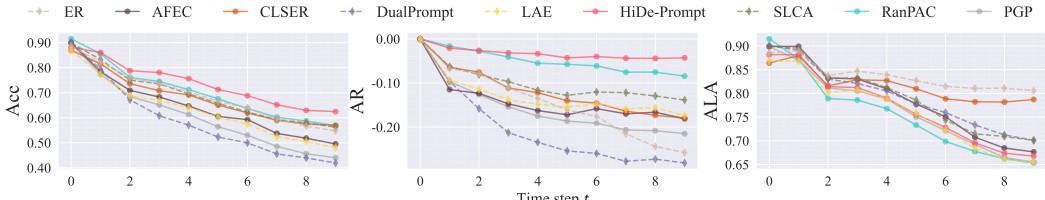

Figure 2: Joint evaluation of the five CL methods (represented by solid lines) using CLDyB with generalization to the other four CL methods (represented by dashed lines).

We present some qualitative visualizations of the CLDyB sequences in Figs. 14 and 18, illustrating that the chosen tasks, originated from various datasets, form sequences in which similar tasks are interspersed with dissimilar ones. We believe this pattern closely resembles real-world CL scenarios with inherent task diversity.

**CLDyB sequences produce reliable evaluation results.** We compare the performance of CL methods across two static CL benchmarks (Krizhevsky, 2009; Hendrycks et al., 2020), CLDyB sequences, and a held-out set—the latter comprising randomly ordered tasks derived from datasets that are excluded from all prior benchmarks. Spearman's rank correlation coefficient (SRCC) and Kendall's rank correlation coefficient (KRCC) are used to assess alignment between each benchmark and the held-out set. From the results in Table 1, we find that the CLDyB sequences achieve the highest correlations, offering a more accurate performance reflection in unseen CL scenarios compared to static benchmarks. Such strong alignments also highlight that CLDyB holds great promise to bridge algorithmic advancements in CL research to real-world applications.

**CLDyB enables multi-dimensional assessment of individual CL methods.** By applying CLDyB separately to each CL method, we seek to measure its worst-case performance, as formalized

in Eqn. (4). Accordingly, we define the Robustness of a CL method as the average Final Acc across multiple CLDyB sequences paired with that method. Additionally, we record the Memory Efficiency[2] in MB, alongside the three standard metrics introduced in Sec. 5.1.

From the results in Fig. 3, we have several interesting observations. First, methods incorporating replay mechanisms, such as CLSER (Arani et al., 2022), HiDe-Prompt (Wang et al., 2023a), SLCA (Zhang et al., 2023), and RanPAC (McDonnell et al., 2023), generally exhibit greater resistance to forgetting, resulting in higher Final Acc. On the downside, these methods tend to be less memory efficient. Second, exemplar-replay methods, specifically ER (Rolnick et al., 2018) and CLSER, demonstrate significantly better model plasticity, whereas parameter-efficient fine-tuning methods like HiDe-Prompt and LAE (Gao et al., 2023) show limited Final ALA, indicating weaker forward transfer capabilities—an aspect often neglected in current CL research. Third, more advanced methods typically outperform their predecessors in terms of performance, but they are not always more robust (*e.g.*, PGP versus DualPrompt and CLSER versus ER). Last, we highlight that while the simple ER does not achieve satisfactory Final Acc and AR, it remains competitive in terms of Final ALA, Robustness, and Memory Efficiency, making it an overall top-ranked CL method.

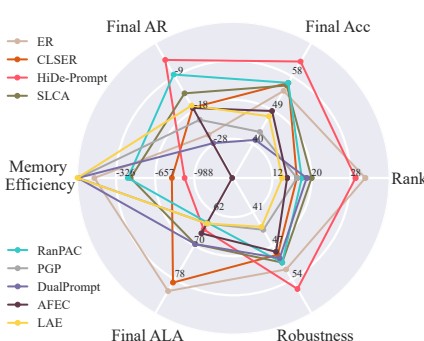

Figure 3: Multi-dimensional assessment of CL methods using CLDyB, with higher values on the axes representing better performance. The comparison highlights their respective strengths and weaknesses, especially how they handle the trade-offs across different dimensions.

### 5.3 FURTHER ANALYSIS

**CLDyB identifies individually challenging task sequences.** In Fig. 4, we visualize the CLDyB sequences for each individual CL method by examining: (a) The Acc trajectories on the commonly challenging CLDyB sequences, and (b) the feature similarity between the selected two tasks, *i.e.*, the average cosine similarity between all possible pairs of image features. This similarity is computed for every task pair within each CLDyB sequence, giving rise to a 2D task-to-task similarity matrix. Each matrix is then flattened into a vector, forming rows of the dendrogram in Fig. 4(b). The clustering patterns in the dendrograms indicate common characteristics shared by certain CL methods. For example, both ER and CLSER rely on exemplar replay, while SLCA and HiDe-Prompt use classifier calibration.

A closer inspection of Fig. 4(b) reveals that the individually challenging CLDyB sequences offer crucial insights into the limitations of different CL methods. Specifically, LAE and AFEC show increased sensitivity to task sequences with significant distribution shifts, as CLDyB tends to select dissimilar tasks for both methods. We hypothesize that this vulnerability may stem from the use of moving average weight ensembles in LAE and parameter regularization in AFEC, both of which are less effective in handling large model updates required for certain CL settings. In contrast, CLSER and ER are more susceptible to sequences of similar tasks, which is consistent with prior theoretical findings, indicating a correlation between task similarity and forgetting (Lee et al., 2021).

**Data pool of CLDyB is expandable over time with AI-generated image data.** The data pool of CLDyB is not intended to remain static. We illustrate this concept by showing how CLDyB can offer valuable feedback for selecting new datasets, enabling the data pool to expand dynamically over time. This continuous evolution helps mitigate the performance saturation issue commonly seen in static benchmarks.

We initially employ CLDyB to evaluate CL methods over three tasks. Based on the analysis of the tree-search history, we observe that tasks related to animals consistently yield higher rewards compared to other tasks with different object categories. This suggests that these animal-related tasks may present particular challenges for the selected CL methods. In response, we employ generative diffusion models, in particular SDXL (Podell et al., 2024), to efficiently expand the data pool,

---

[2]Equivalent to the negative memory storage (detailed in Appendix B.3).

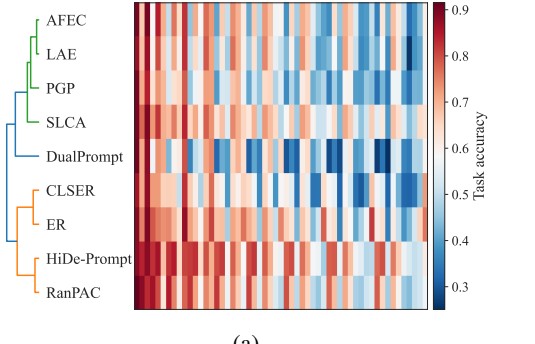 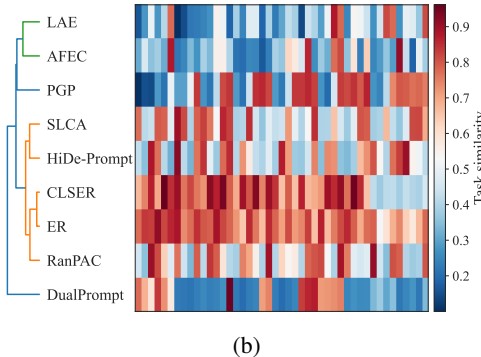

(a)                                                    (b)

Figure 4: Dendrograms of CL methods. (a) Acc trajectories on the commonly challenging CLDyB sequences. (b) Stack of flattened versions of the 2D task-to-task similarity matrices obtained on individually challenging CLDyB sequences. The task similarity values are normalized by $s(\cdot)$ in Eqn. (5) to the range $[0, 1]$ for improved visualization. Both dendrograms exhibit noticeable consistency in their hierarchical structures, reflecting commonality in CL methods.

adding generated images of novel animal categories. In Fig. 5, we compare the performance of the CL methods on upcoming tasks selected by CLDyB using both the original and the augmented data pools.

We find that the AI-generated classes are indeed frequently selected, indicating their prominence in the CLDyB sequences (see Fig. 21). This leads to a noticeable drop in Final AA and AR, which verifies the added difficulty of the augmented data pool relative to the original, and confirms that our data pool is readily expandable to resolve performance saturation. Additionally, the tasks depicted in Fig. 21 demonstrate variations in image styles, such as shifts from photographic images to sketches and paintings, pointing out potential difficulty faced by current CL methods in adapting to stylistic changes over time. In particular, rehearsal-free methods like AFEC (Wang et al., 2021) and PGP (Qiao et al., 2024) exhibit more substantial performance degradations when encountering task sequences with style transitions.

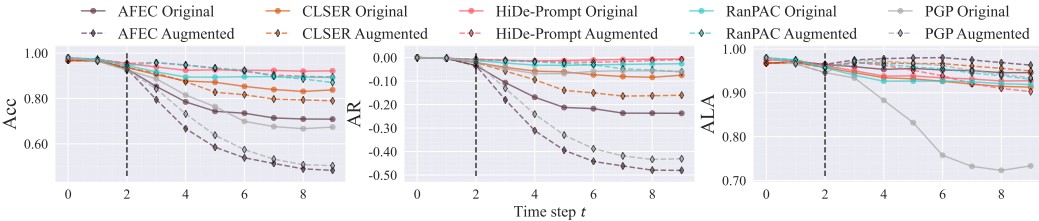

Figure 5: Performance comparison of CL methods on upcoming tasks selected by CLDyB from the original and the augmented data pools. Additional diffusion-generated images are added to the data pool at time step $t = 2$ (denoted by the dashed line).

## 5.4 ABLATION STUDIES

To isolate the contributions of individual components in CLDyB, we systematically remove each of them, resulting in four degenerates: 1) Sampling tasks randomly (as a performance lower bound), 2) replacing greedy task sampling with uniform sampling from the classes within each dataset in the data pool for candidate task set construction, 3) disabling functional task clustering, and 4) substituting MCTS for selecting the next task with a method that consistently chooses tasks most similar in representation to previous tasks, as suggested in (Bell & Lawrence, 2022).

In Fig. 6, it is evident that the full version of CLDyB outperforms all degenerates in identifying more challenging task sequences, which leads to lower ACC and AR, thus validating the necessity of each component. Furthermore, to assess the effectiveness of CLDyB in handling long task sequences, we

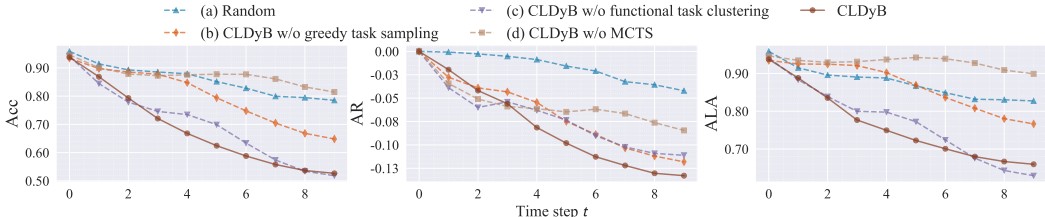

Figure 6: Average performance comparison of the nine CL methods on the CLDyB sequences and those generated by four CLDyB degenerates.

conduct experiments on an extended sequence consisting of 40 tasks, with results shown in Fig. 9. It is clear that, on average, CL methods consistently underperform on CLDyB sequences compared to random sequences. Finally, Fig. 10 illustrates that CLDyB remains effective in dynamically identifying commonly challenging sequences to combat data contamination and performance saturation using alternative foundation models (*i.e.*, CLIP-ViT-Base (Radford et al., 2021)).

## 6 CONCLUSION AND DISCUSSION

We have introduced CLDyB, a dynamic computational benchmarking framework that systematically identifies challenging and algorithm-dependent task sequences based on MDPs. To make sequential task generation computationally feasible, CLDyB makes two reasonable approximations: action space reduction via greedy task sampling and clustering, and value function approximation via MCTS. By evaluating nine state-of-the-art CL methods, we demonstrated that CLDyB sequences expose critical weaknesses in existing methods while providing reliable evaluations aligned with real-world generalization. CLDyB represents a paradigm shift in evaluating CL methods. By transitioning from static benchmarks to dynamic and algorithm-dependent evaluation protocols, we pave the way for CL methods that genuinely adapt to open-world complexity. We hope that CLDyB and its generated task sequences provide a foundation for developing next-generation CL methods that strike a better balance between stability and plasticity.

While CLDyB represents a significant step towards reliable and realistic CL benchmarking, several promising directions remain for future research. First, one of our primary motivations for CLDyB is the issue of data contamination. However, quantifying data contamination precisely is challenging due to inaccessible pre-training data and the need to assess its overlap with downstream tasks. While heuristic approaches (*e.g.*, task difficulty metrics) offer approximations, they may conflate contamination with other factors influencing difficulty, limiting their reliability. Although empirical studies show that CL performance becomes inflated when pre-training and CL tasks overlap significantly, defining a threshold of contamination beyond which CL evaluations lose validity remains difficult. Addressing this could refine benchmark design and strengthen evaluation protocols for CL methods. Second, the development of self-evolving data pools through human-AI collaboration and generative models presents an important avenue. Although we demonstrated preliminary success using diffusion-generated image data by further exploiting known failure scenarios, a systematic approach is needed to automatically and dynamically expand tasks in the data pool in terms of quality, difficulty, and diversity. Third, the current MCTS-based method, while effective, offers only a coarse approximation of the value function and incurs high computational costs. Hybrid methods that integrate learned value estimators—without the need for computationally expensive continual model adaptation—alongside tree search could improve scalability. Fourth, although our experiments have thus far focused on image recognition, CLDyB is readily extendable to other domains, including video, text, and embodied AI. Last, establishing a community-driven platform for the collaborative evolution of CLDyB-enabled benchmarks is highly desirable. Inspired by Dynabench (Kiela et al., 2021), a web-based interface could facilitate the submission of new datasets to expand the data pool and accommodate new CL models while ensuring backward compatibility for fair and consistent progress tracking.

## ACKNOWLEDGEMENTS

We would like to thank Xuelin Liu for assistance with the plots and diagrams. This work was supported in part by the Hong Kong RGC General Research Fund (11220224), the CityU Applied Research Grant (9667264), and an Industry Gift Fund (9229111).

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

APPENDIX

# A  CLDYB DATA POOL

## A.1  CLDYB DATA POOL CONSTRUCTION

**Photographic image data.** This portion of the CLDyB data pool includes $2,043$ classes from $22$ publicly available image recognition datasets, and can be categorized into two major domains:

- **Natural and biological scenes.** This category includes images of fauna, flora, and various natural elements and ecosystems, spanning multiple levels of granularity, from coarse-grained categories (*e.g.*, different animal species in Animal-90 (Banerjee, 2024)) to fine-grained distinctions (*e.g.*, specific butterfly species in Butterfly-70 (Thai, 2024)).
- **Man-made scenes.** This category contains images depicting clothing, food, building, and entertainment scenes, capturing a rich diversity of cultural and historical contexts from multiple geographic regions and time periods. For example, it includes Food-101 (Bossard et al., 2014), CNFOOD-241(Aluza, 2024) with Chinese cuisine, and Indian Food Images (Banerjee, 2023), which together represent a variety of regional culinary traditions.

**Diffusion-generated image data.** In addition to photographic images, our data pool incorporates four AI-generated image datasets by SDXL (Podell et al., 2024) to expand its data distribution and introduce new challenges for CL. These datasets contain 360 classes across three animal categories and one product category, created using the following prompts:

- '*A high-quality image of a kind of animal: {animal name}*'
- '*A high-quality sketch image of a kind of animal: {animal name}*'
- '*A high-quality {image style} image of a kind of animal: {animal name}*'
- '*A high-quality image of a kind of product: {product name}*'

Table 2 gives the details of these datasets, which retain the licensing terms of their original sources whenever explicitly stated. In cases where no specific license is provided, the dataset is distributed under the CC BY-NC 4.0 license[3], permitting use only for non-commercial purposes.

## A.2  CURRENT CL BENCHMARKS

The standard practice for evaluating class-incremental learning methods, first established in (Rebuffi et al., 2016), involves partitioning the classes of a labeled dataset into sequentially ordered tasks with non-overlapping class labels for training and testing. Commonly used datasets include MNIST (Deng, 2012), CIFAR-100 (Krizhevsky, 2009), CUB-200 (Wah et al., 2011), and various ImageNet derivatives (Le & Yang, 2015; Hendrycks et al., 2020). Additionally, benchmarks like OmniBenchmark and VTAB (Zhou et al., 2023a) aggregate multiple standard classification datasets, each regarded as an individual task. Beyond remixing available datasets, several new datasets have been specifically curated, including Core50 (Lomonaco & Maltoni, 2017), which features diverse objects in varying contexts, and CLEAR (Lin et al., 2021), which captures natural temporal progressions of visual concepts. Furthermore, Liao et al. (2023) introduced CGQA and COBJ benchmarks to assess the compositional generalization abilities of CL methods. However, existing CL benchmarks remain static, and do not explicitly account for assessment of CL methods with strong foundation models.

---

[3]https://creativecommons.org/licenses/by-nc-sa/4.0/

Table 2: Statistics of the CLDyB data pool. For certain datasets, the number of classes (*i.e.*, # of classes) is smaller than the original, as we have selected only those with fewer than 45 images.

| Type | Dataset | # of classes | # of images |
|---|---|---|---|
| Photographic image data | Fruits-Vegetable (Seth, 2024) | 35 | 3,047 |
| | Jute-Pest (Islam, 2024) | 17 | 7,151 |
| | Animal-90 (Banerjee, 2024) | 90 | 5,400 |
| | Sea-Animal (Lanz, 2024) | 23 | 13,711 |
| | Rock (Technologies, 2024) | 9 | 3,687 |
| | Butterfly-75 (Thai, 2024) | 75 | 6,499 |
| | Clothing-20 (Kaiska, 2024) | 17 | 5,325 |
| | Apparel (Grigorev, 2024) | 37 | 16,170 |
| | Food101 (Bossard et al., 2014) | 101 | 101,000 |
| | FoodX251 (Kaur et al., 2019) | 250 | 118,441 |
| | Indian-Food (Banerjee, 2023) | 80 | 4,000 |
| | CNFOOD241 (Aluza, 2024) | 240 | 170,835 |
| | Oxford5k (Philbin et al., 2007) | 17 | 5,063 |
| | SUN397 (Xiao et al., 2010) | 339 | 17,355 |
| | Places365 (Zhou et al., 2017) | 365 | 1,803,460 |
| | RESISC45 (Cheng et al., 2017) | 45 | 5,100 |
| | EuroSAT-RGB (Helber et al., 2018) | 10 | 27,000 |
| | MLRSNet (Qi et al., 2020) | 46 | 109,161 |
| | FGVC-Aircraft (Maji et al., 2013) | 100 | 10,000 |
| | Balls30 (Piosenka, 2024a) | 29 | 3,708 |
| | Sports100 (Piosenka, 2024b) | 100 | 13,492 |
| | One-Piece-Anime (Sisodiya, 2024) | 18 | 11,737 |
| Diffusion-generated image data | Product-71 | 71 | 7,095 |
| | Animal-Multi-Style | 133 | 19,962 |
| | Animal-Sketch | 77 | 8,494 |
| | Animal | 79 | 8,292 |
| CLDyB data pool | Combined | 2,403 | 2,505,185 |

## B    CL METHODS

### B.1    TRADITIONAL CL METHODS

Traditional class-incremental learning methods can be broadly divided into five categories: 1) Replay-based, 2) regularization-based, 3) model-based, 4) optimization-based, and 5) representation-based approaches. Specifically,

- **Replay-based methods** use a memory buffer to store and replay previous samples when learning new tasks, with recent emphasis on exemplar selection (Aljundi et al., 2019; Yoon et al., 2021), constraint optimization (Lopez-Paz & Ranzato, 2017; Chaudhry et al., 2018), and generative replay (Shin et al., 2017).

- **Regularization-based methods** (Kirkpatrick et al., 2017) penalize significant changes in CL methods to preserve acquired knowledge. These include regularization of important parameters, where the importance is measured by the Fisher information matrix (Kirkpatrick et al., 2017), weight uncertainty (Ahn et al., 2019), and variational posterior (Nguyen et al., 2018), as well as functional regularization that leverages Gaussian processes (Titsias et al., 2020), knowledge distillation (Li & Hoiem, 2016; Buzzega et al., 2020), and contrastive learning (Cha et al., 2021).

- **Model-based methods** often involve model expansion (Schwarz et al., 2018; Wang et al., 2023a) and parameter isolation (Wang et al., 2022) to manage task-specific knowledge.

- **Optimization-based methods** employ techniques like orthogonal gradient projections (Farajtabar et al., 2019; Saha et al., 2021) and meta-learning (Riemer et al., 2018; Gupta et al., 2020) to mitigate negative interference between tasks.

- **Representation-based methods** exploit the advantages of meta-learning (Javed & White, 2019), self-supervised learning (Gallardo et al., 2021), and large-scale pre-training (Mehta et al., 2023) to improve feature representations in model initialization and adaptation.

We refer interested readers to (Wang et al., 2023b) for a comprehensive survey.

## B.2 SELECTED CL METHODS FOR CLDYB

We provide a brief description of the selected CL methods for CLDyB. All CL methods work seamlessly with foundation models of varying backbones. Table 3 summarizes the design principles of different CL methods for better comparison.

- **ER** (Rolnick et al., 2018) is the simplest replay-based method that stores previous exemplars in a fixed memory buffer, and replays them during training to mitigate catastrophic forgetting.
- **AFEC** (Wang et al., 2021) enhances elastic weight consolidation (Kirkpatrick et al., 2017) by introducing a second Fisher information matrix derived from the current and task-specific fine-tuned models as regularization. Inspired by biological neural networks, this approach enables selective forgetting of old knowledge conflicting with new experiences, moving beyond rigid retention strategies.
- **CLSER** (Arani et al., 2022) employs two models: a stable one (for long-term memory) and a plastic one (for short-term memory), updated via distinct exponential moving average (EMA) strategies. During training, it incorporates the mean squared error between the working model and the most confident EMA model as a form of distillation loss.
- **DualPrompt** (Wang et al., 2022), a prompt-based CL method, improves upon L2P (Zhou et al., 2021) by attaching prompts to multiple attention layers. It also distinguishes task-agnostic and task-specific prompts, enhancing the model's ability to learn both global (cross-task) and local (within-task) knowledge.
- **LAE** (Gao et al., 2023) maintains an online parameter-efficient module through regular training and updates an offline parameter-efficient module using EMA. During inference, it employs a routing strategy by selecting the more confident prediction from the two modules.
- **HiDe-Prompt** (Wang et al., 2023a) enhances prior prompt-based methods with several improvements, including a prompt-ensemble strategy, a contrastive feature constraint, and a joint optimization of three tasks: within-task prediction, task-identity prediction, and task-adaptive prediction.
- **SLCA** (Zhang et al., 2023) employs a dual learning rate strategy: the feature encoder (backbone) is fine-tuned with a smaller learning rate to maintain stable feature extraction, while the classifier is updated with a larger learning rate. Additionally, SLCA mitigates classifier forgetting by modeling and replaying class-wise feature distributions, ensuring robust calibration of the classifier over time.
- **PGP** (Qiao et al., 2024) demonstrates that orthogonalizing prompt gradients via singular value decomposition prevents forgetting.
- **RanPAC** (McDonnell et al., 2023) freezes the backbone after initial task training, and applies non-learnable random projections to enhance feature separability. It further decorrelates class prototypes using the inverse of the Gram matrix of the projected features, achieving strong performance without the use of any memory buffer.

Table 3: Summary of design principles of the nine selected CL methods for CLDyB.

| Method | Replay | | Training | | Regularization | | | Model | |
|---|---|---|---|---|---|---|---|---|---|
| | Exemplar | Statistic | Full | Parameter-efficient | Functional-based | Parameter-based | Orthogonality-based | Isolation or expansion | Ensemble |
| ER | ✓ | | ✓ | | | | | | |
| AFEC | | | ✓ | | | ✓ | | | |
| CLSER | ✓ | | ✓ | | ✓ | | | | ✓ |
| DualPrompt | | | | ✓ | | | | ✓ | |
| LAE | | | | ✓ | | | | | ✓ |
| HiDe-Prompt | | ✓ | | ✓ | | | | ✓ | ✓ |
| SLCA | | ✓ | ✓ | | | | | | |
| RanPAC | | ✓ | ✓ | | | | | | |
| PGP | | | | ✓ | | | ✓ | ✓ | |

## B.3 MEMORY FOOTPRINT CALCULATION

In the calculation of memory footprint, we consider three main sources beyond the ViT backbone: 1) learner parameters including model replicas (*e.g.*, old and EMA-updated models) and parameter-

efficient modules, 2) dataset metadata containing per-task statistics like mean and variance, and 3) image storage retaining raw 8-bit $224 \times 224$ RGB exemplars from previous tasks. All data are stored in 32-bit floats.

## C  PSEUDOCODE FOR CLDYB

We present the pseudocode for CLDyB in Algorithms 1, 2, and 3.

---

**Algorithm 1** $\tilde{\mathcal{A}}_t = \texttt{GreedyTaskSampling}(\mathcal{D}_t, \mathcal{F}_{t-1}, \tilde{B}, K)$

---

**Require:** Data pool $\mathcal{D}_t$, continually learned classifiers $\mathcal{F}_{t-1} = \left\{ f_{t-1}^{(i)} \right\}_{i=1}^{M}$, # of candidate tasks to sample $\tilde{B}$, and # of classes per task $K$

**Return:** $\tilde{\mathcal{A}}_t$                                                  ▷ Reduced action space

1: $\Psi(\mathcal{C}_u, \mathcal{C}_v) = \frac{1}{M} \sum_{i=1}^{M} s\left( \text{CosSim}\left( \boldsymbol{u}_t^{(i)}, \boldsymbol{v}_t^{(i)} \right) \right), \forall u, v \in \mathcal{D}_t$   ▷ Compute pairwise potentials

2: $\tilde{\mathcal{A}}_t \leftarrow \emptyset$

3: **for** $j \in \{1, \ldots, \tilde{B}\}$ **do**                                        ▷ Sample each task

4:     $\mathcal{V}_0^\star \leftarrow \emptyset$

5:     **for** $k \in \{1, \ldots, K\}$ **do**                              ▷ Sample each class in the task

6:         **if** $k == 1$ **then**                                     ▷ Sample the first class uniformly

7:             $u_k^\star \sim [|\mathcal{D}_t|]$

8:         **else**                                         ▷ Sample subsequent classes according to Eqn. (6)

9:             $u_k^\star = \arg\max_u \prod_{v \in \mathcal{V}_{k-1}^\star} \Psi(\mathcal{C}_u, \mathcal{C}_v)$

10:         **end if**

11:         $\mathcal{V}_k^\star \leftarrow \mathcal{V}_{k-1}^\star \bigcup u_k^\star$

12:     **end for**

13:     $\tilde{\mathcal{A}}_t \leftarrow \tilde{\mathcal{A}}_t \bigcup \mathcal{V}_K^\star$                              ▷ Add the task to the candidate set

14: **end for**

---

**Algorithm 2** $\bar{\mathcal{A}}_t = \texttt{FunctionalTaskClustering}(\tilde{\mathcal{A}}_t, \mathcal{F}_{t-1}, \bar{B}, C)$

---

**Require:** Candidate task set $\tilde{\mathcal{A}}_t$, continually learned classifiers $\mathcal{F}_{t-1} = \left\{ f_{t-1}^{(i)} \right\}_{i=1}^{M}$, # of candidate tasks to sample $\bar{B}$, and # of task clusters $C$

**Return:** $\bar{\mathcal{A}}_t$                                                 ▷ Condensed action space

1: $\mathbf{G} \leftarrow \mathbf{0}^{|\tilde{\mathcal{A}}_t| \times M}$                                      ▷ Initialize the task feature matrix

2: **for** $i \in \left\{ 1, \ldots, \left| \tilde{\mathcal{A}}_t \right| \right\}$ **do**

3:     **for** $j \in \{1, \ldots, M\}$ **do**          ▷ Compute the average negative log-likelihood of sample predictions for task $\tilde{\mathcal{T}}_t^{(i)}$ using the $j$-th $k$-NN classifier $p_{t-1}^{(j)}$ built from $f_{t-1}^{(j)}$

4:         $\mathbf{G}[i, j] = -\frac{1}{\left| \tilde{\mathcal{T}}_t^{(i)} \right|} \sum_{(\boldsymbol{x}, y) \in \tilde{\mathcal{T}}_t^{(i)}} \log p_{t-1}^{(j)}(y|\boldsymbol{x})$

5:     **end for**

6: **end for**

7: $\left\{ \bar{\mathcal{T}}_t^{(1)}, \bar{\mathcal{T}}_t^{(2)}, \ldots, \bar{\mathcal{T}}_t^{(C)} \right\} = \texttt{KMeans}(\mathbf{G}, C)$                  ▷ Cluster $\tilde{\mathcal{A}}_t$ into $C$ task clusters

8: $\bar{\mathcal{A}}_t \leftarrow \emptyset$

9: **for** $i \in \{1, 2, \ldots, \bar{B}\}$ **do**                              ▷ Sample each task without replacement

10:     $j \sim [C]$                                            ▷ Sample one task cluster uniformly

11:     $\tilde{\mathcal{T}}_t \sim \bar{\mathcal{T}}_t^{(j)}$                                 ▷ Sample one task from the cluster uniformly

12:     $\bar{\mathcal{T}}_t^{(j)} \leftarrow \bar{\mathcal{T}}_t^{(j)} \setminus \tilde{\mathcal{T}}_t$                                     ▷ Remove the task from the cluster

13:     $\bar{\mathcal{A}}_t \leftarrow \bar{\mathcal{A}}_t \bigcup \tilde{\mathcal{T}}_t$                                       ▷ Add the task to the candidate set

14: **end for**

---

---

**Algorithm 3** CLDyB

---

**Require:** Data pools $\{\mathcal{D}_t\}_{t=1}^N$, foundation models $\mathcal{F}_0 = \left\{ f_0^{(i)} \right\}_{i=1}^M$, CL methods $\{\text{CL}^{(i)}\}_{i=1}^M$, # of candidate tasks $\tilde{B}$ and $\bar{B}$, # of task clusters $C$, and # of classes in each task $K$

**Return:** $\mathcal{T}_{<N+1}$        ▷ CLDyB task sequence

1: $\mathcal{T}_{<1} \leftarrow \emptyset$
2: **for** $t = \{1, \ldots, N\}$ **do**
3:     $\tilde{\mathcal{A}}_t = \text{GreedyTaskSampling}(\mathcal{D}_t, \mathcal{F}_{t-1}, \tilde{B}, K)$      ▷ Algorithm 1
4:     $\bar{\mathcal{A}}_t = \text{FunctionalTaskClustering}(\tilde{\mathcal{A}}_t, \mathcal{F}_{t-1}, \bar{B}, C)$      ▷ Algorithm 2
5:     $\mathcal{T}_t = \text{MCTS}(\text{current state} = \mathcal{F}_{t-1}, \text{action space} = \bar{\mathcal{A}}_t, \text{value function} = J_{\mu_t}(\mathcal{F}_{t-1}))$   ▷
    Identify the optimal task according to Eqn. (7)
6:     $\mathcal{F}_t \leftarrow \{f_t^{(i)}\}_{i=1}^M$, where $f_t^{(i)} = \text{CL}^{(i)}(f_{t-1}^{(i)}, \mathcal{T}_t)$   ▷ Train each classifier on the selected task
7:     $\mathcal{T}_{<t+1} \leftarrow \mathcal{T}_{<t} \bigcup \mathcal{T}_t$         ▷ Add the task to the CLDyB sequence
8: **end for**

---

# D   ADDITIONAL RESULTS

## D.1   CLDYB SEQUENCES WITH VARYING DIFFICULTY LEVELS

While the main text describes CLDyB as selecting the hardest task at each time step (via greedy maximization of Eqn. (7)), we extend it to generate sequences with varying difficult levels. This can be done by replacing greedy task sampling with task reward-based sampling:

$$p\left(\tilde{\mathcal{T}}_t\right) = \frac{\exp\left(J_{\tilde{\mathcal{T}}_t}(\mathcal{F}_{t-1})/\tau\right)}{\sum_{\mathcal{T}'_t \in \bar{\mathcal{A}}_t} \exp\left(J_{\mathcal{T}'_t}(\mathcal{F}_{t-1})/\tau\right)}, \tag{8}$$

where we slightly abuse the math notation by putting the action (*i.e.*, $\tilde{\mathcal{T}}_t$ or $\mathcal{T}'_t$) in the subscript of the value function $J$ instead of the policy $\mu_t$ to be optimized. The temperature hyperparameter $\tau$ controls the difficulty bias. Lower $\tau$ values ($\tau \to 0$) prioritize high-reward (challenging) tasks, mimicking greedy sampling, while higher $\tau$ values promote diversity by flattening the probability distribution. All other components in CLDyB remain unchanged.

Using reward-based sampling within CLDyB, we generate two additional sets of sequences at the medium and easy difficulty levels, retaining the original as hard sequences. Table 4 presents the results, where we observe two key trends. First, performance improves predictably as difficulty decreases, with the hard sequences providing performance lower bounds. Second, while minor ranking fluctuations occur, results across all CLDyB sequences remain strongly correlated with those on held-out sequences.

Table 4: Final average accuracy (%) of CL methods on CLDyB sequences at the easy, medium, and hard levels.

| Method | Easy | Medium | Hard | Held-out |
|---|---|---|---|---|
| ER (Rolnick et al., 2018) | 76.3 | 60.5 | 54.8 | 79.9 |
| AFEC (Wang et al., 2021) | 72.5 | 51.9 | 49.5 | 76.3 |
| CLSER (Arani et al., 2022) | 75.4 | 60.7 | 57.0 | 81.1 |
| DualPrompt (Wang et al., 2022) | 69.0 | 53.7 | 41.9 | 70.8 |
| LAE (Gao et al., 2023) | 70.1 | 50.9 | 48.1 | 71.1 |
| HiDe-Prompt (Wang et al., 2023a) | 80.1 | 67.6 | 62.5 | 84.9 |
| SLCA (Zhang et al., 2023) | 77.2 | 59.7 | 56.2 | 80.3 |
| RanPAC (McDonnell et al., 2023) | 77.0 | 62.2 | 56.9 | 81.0 |
| PGP (Qiao et al., 2024) | 69.2 | 52.0 | 44.0 | 68.7 |
| SRCC (per column & held-out) | 0.867* | 0.833* | 0.983* | N/A |
| KRCC (per column & held-out) | 0.722* | 0.667* | 0.944* | N/A |

## D.2   ADDITIONAL FIGURES

Figs 7 to 12 provide additional results under various conditions. Specifically,

- Fig. 7 evaluates CL methods on random sequences.
- Fig. 8 compares the commonly challenging CLDyB sequences with random sequences.
- Fig. 9 tests the robustness of CLDyB in long-sequence, commonly challenging scenarios.
- Fig. 10 validates CLDyB with CLIP-ViT-Base (Radford et al., 2021) in commonly challenging scenario, a stronger foundation model pre-trained on a much larger web-scale dataset, FYCC-100M (Thomee et al., 2016), in comparison to ViT-Base-Sup21K (Dosovitskiy et al., 2021).
- Fig. 11 assesses CL methods on the commonly challenging CLDyB sequences, when combining ViT-Base-Sup21K (Dosovitskiy et al., 2021) and CLIP-ViT-Base (Radford et al., 2021).
- Fig. 12 benchmarks CL methods separately using the hybrid configuration from Fig. 11.
- Fig. 13 compares CL methods on the CLDyB sequences, discovered from the original data pool and its augmented version with AI-generated image data, respectively.

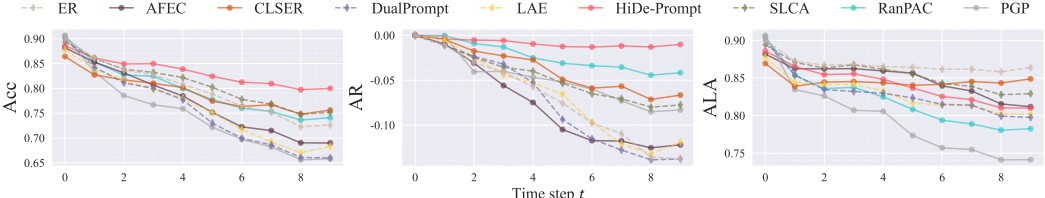

Figure 7: Performance comparison of CL methods on random sequences drawn from the CLDyB data pool.

### D.3 VISUALIZATION OF CLDyB SEQUENCES

Figs 14 to 17 show t-SNE visualizations (van der Maaten & Hinton, 2008) of tasks in the commonly and individually challenging CLDyB sequences. Figs. 18 to 20 provide sample images from the commonly challenging CLDyB sequences. Additionally, Fig 21 demonstrates the effect of adding AI-generated image data to the CLDyB data pool over time.

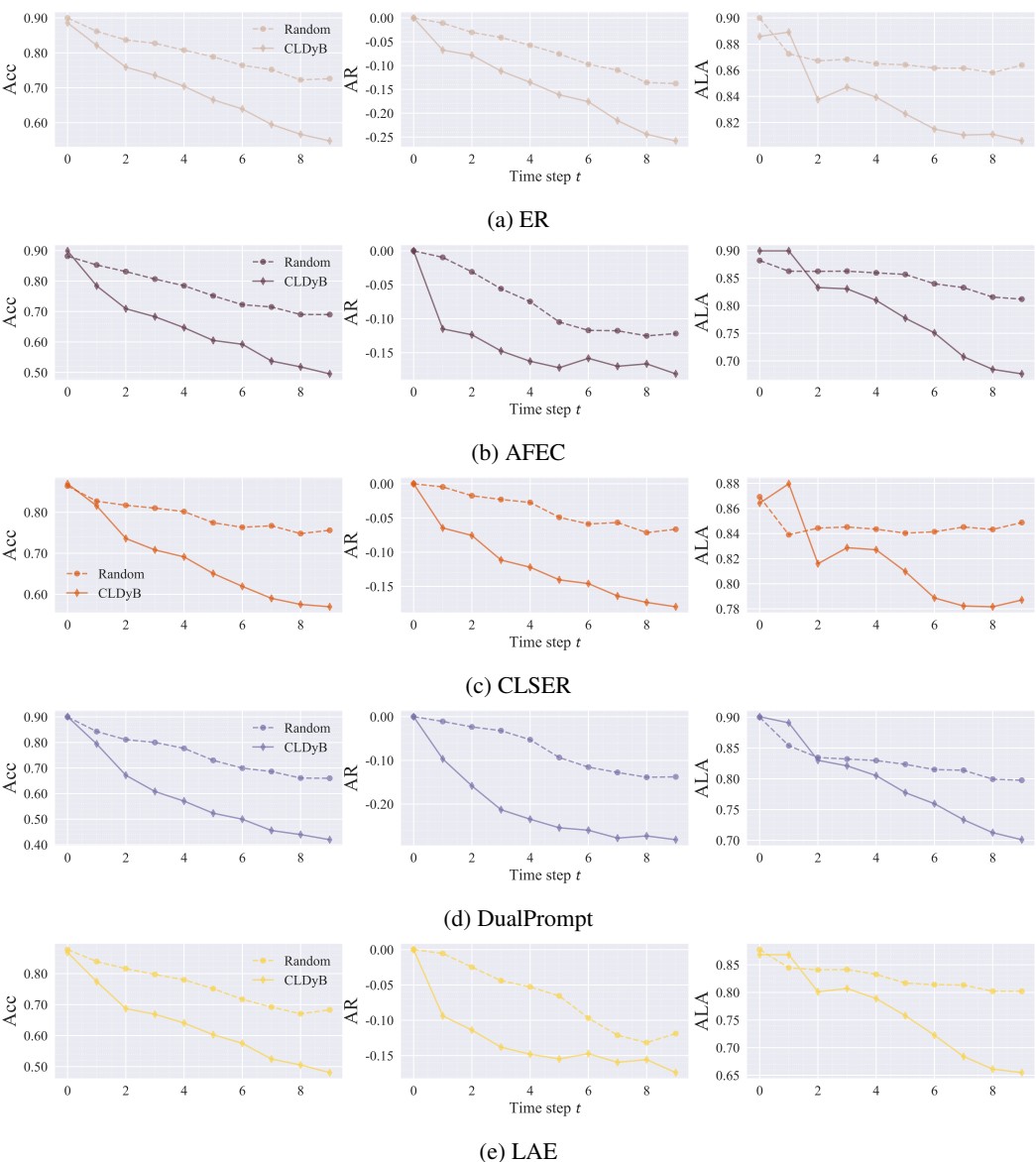

Figure 8: Joint evaluation of the five CL methods using CLDyB relative to random sequences. All methods use the ViT-Base backbone (Dosovitskiy et al., 2021), pre-trained on ImageNet-21K.

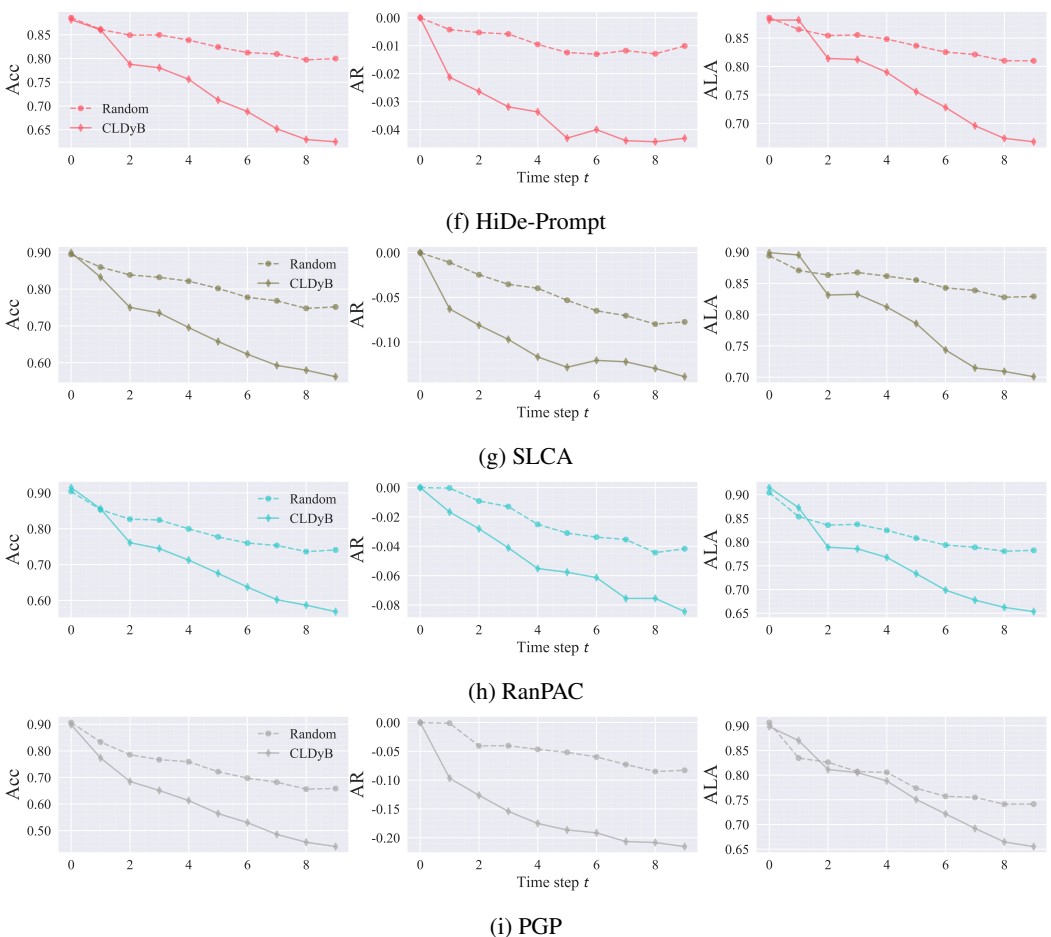

Figure 8: Continued. Independent testing of the four CL methods using the same CLDyB sequences as in Figs. 8(a)-(e), relative to random sequences.

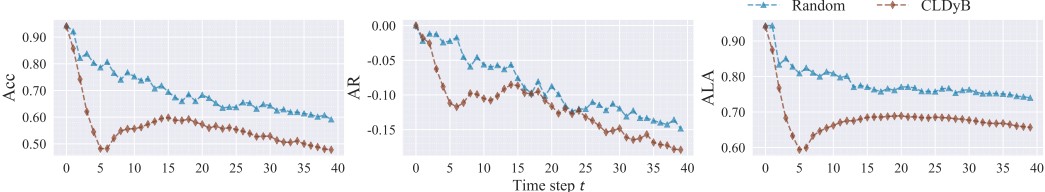

Figure 9: Average performance comparison of CL methods on random and long commonly challenging CLDyB sequences (when $N = 40$).

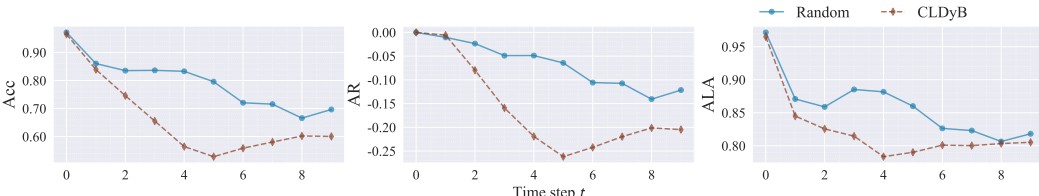

Figure 10: Average performance comparison of CL methods on random and CLDyB sequences, with CLIP-ViT-Base (Radford et al., 2021) as the foundation model.

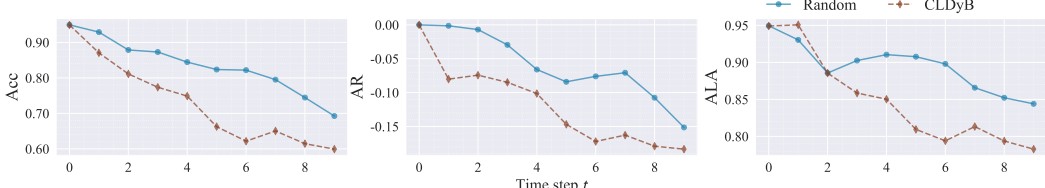

Figure 11: Average performance comparison of CL methods on random and CLDyB sequences, using a mixture of ViT-Base-Sup21K (Dosovitskiy et al., 2021) and CLIP-ViT-Base (Radford et al., 2021).

(a) ER

(b) AFEC

(c) CLSER

(d) DualPrompt

(e) LAE

Figure 12: Performance comparison of individual CL methods on random and commonly challenging CLDyB sequences.

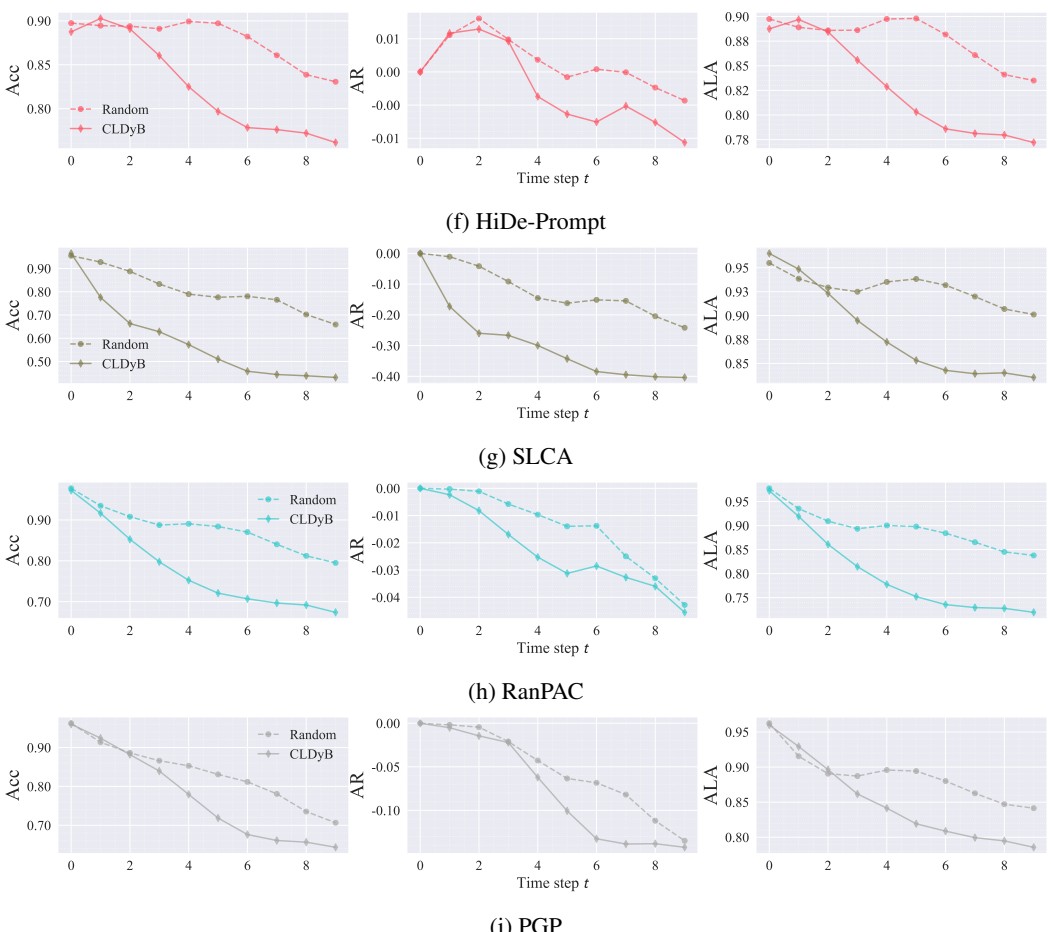

Figure 12: Continued. Performance comparison of individual CL methods on random and commonly challenging CLDyB sequences.

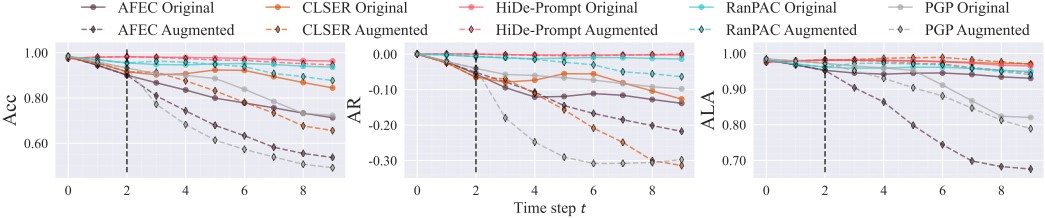

Figure 13: Performance comparison of CL methods on upcoming tasks selected by CLDyB from the original and the augmented data pools. Additional diffusion-generated images are added to the data pool at time step $t = 2$ (denoted by the dashed line).

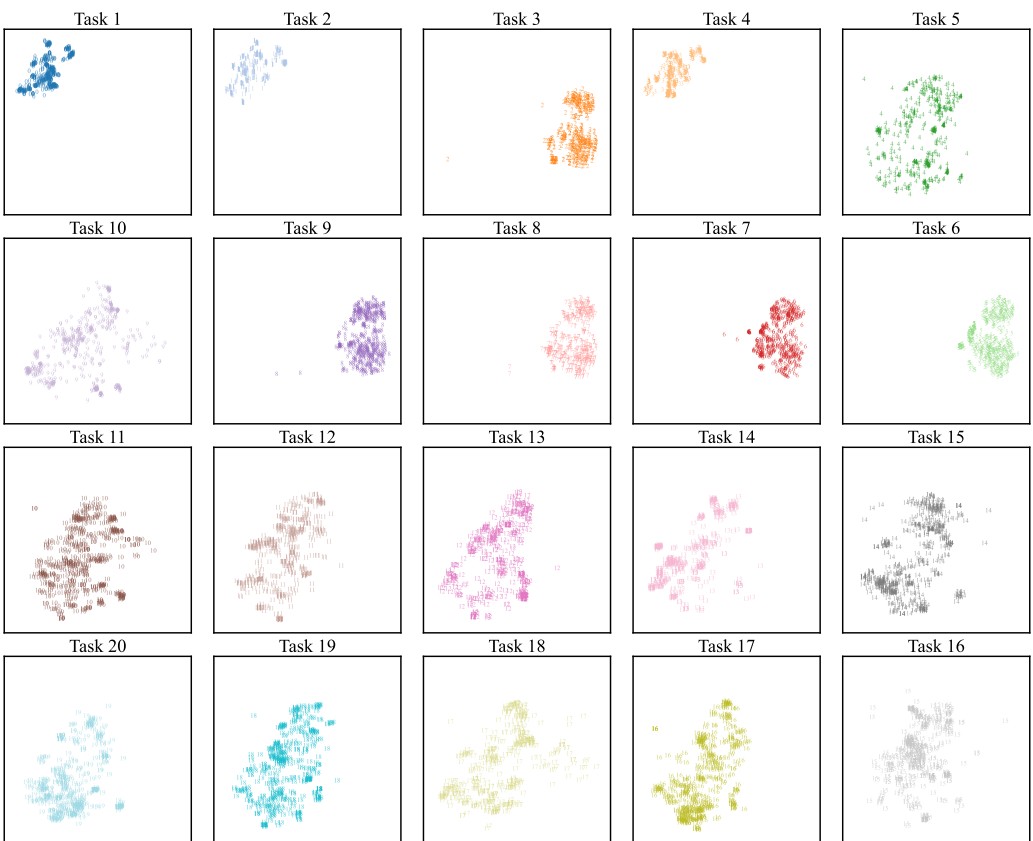

Figure 14: t-SNE visualizations of tasks in the commonly challenging CLDyB sequences.

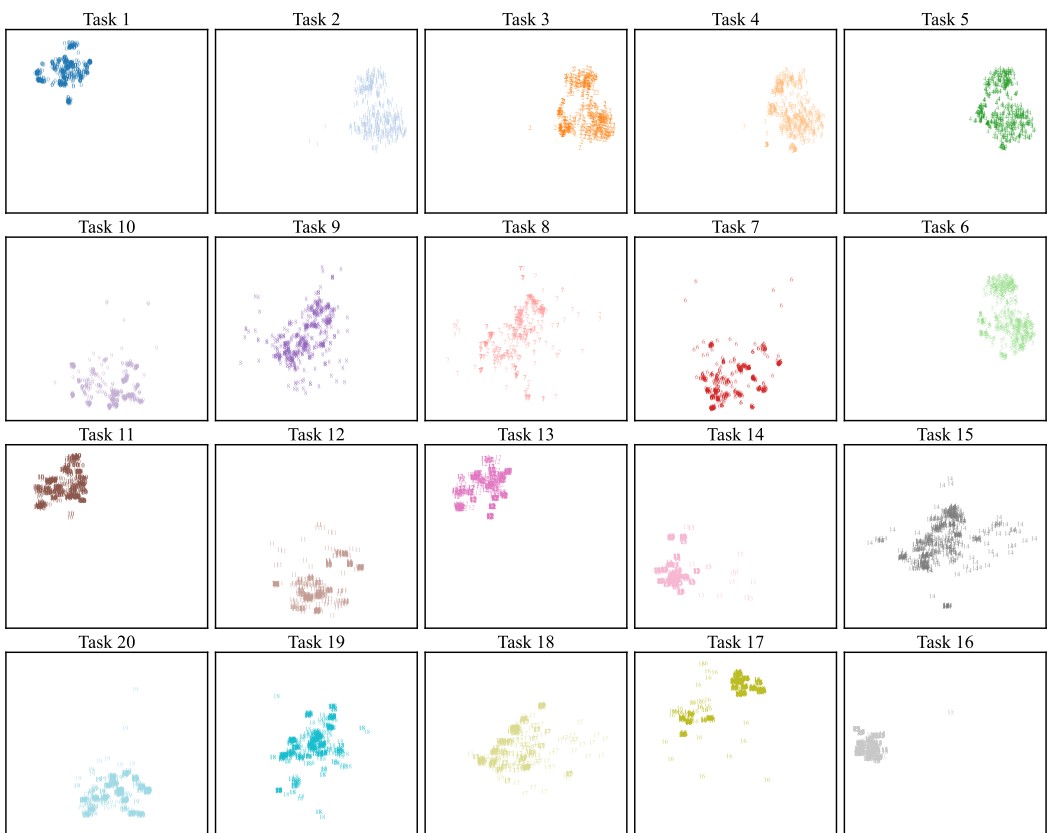

Figure 15: t-SNE visualizations of tasks in the individually challenging CLDyB sequences for ER.

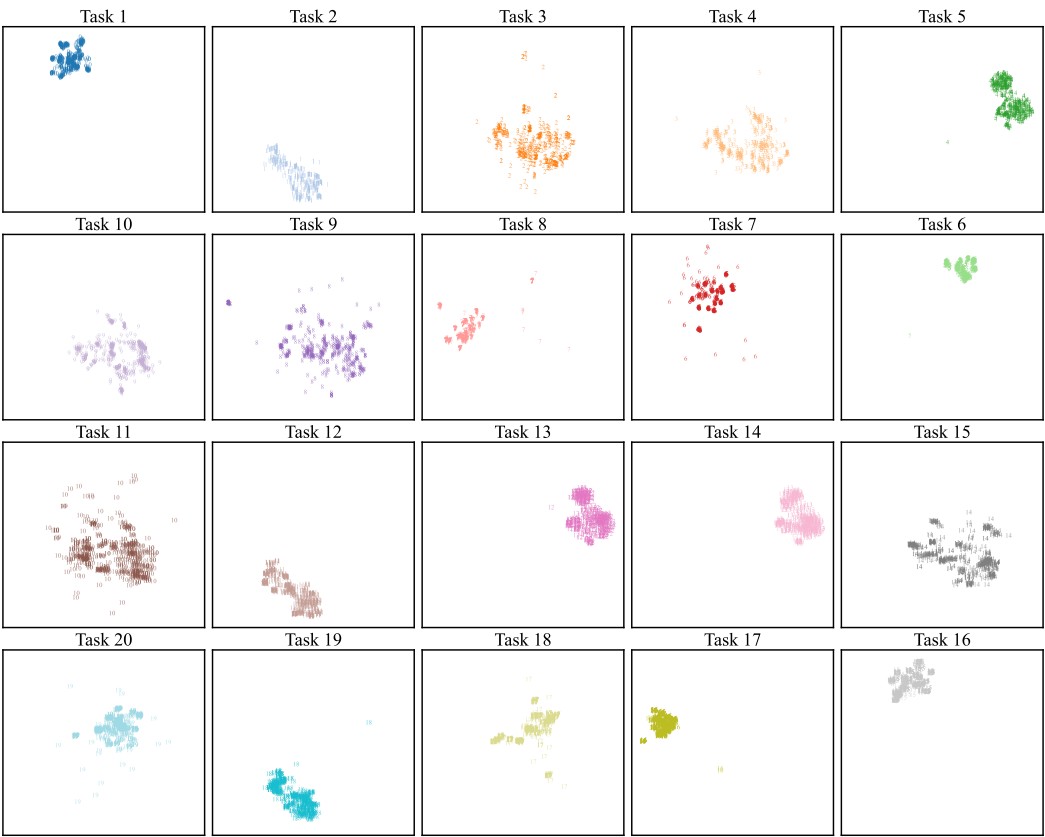

Figure 16: t-SNE visualizations of tasks in the individually challenging CLDyB sequences for LAE.

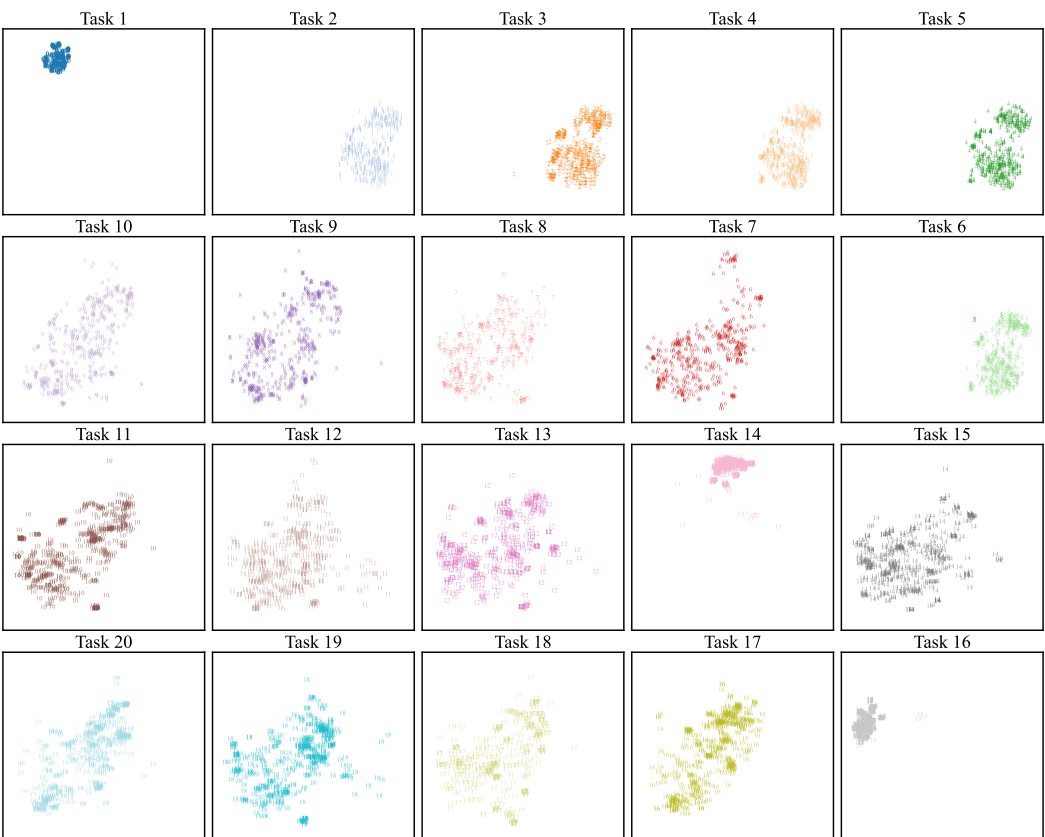

Figure 17: t-SNE visualizations of tasks in the individually challenging CLDyB sequences for Ran-PAC.

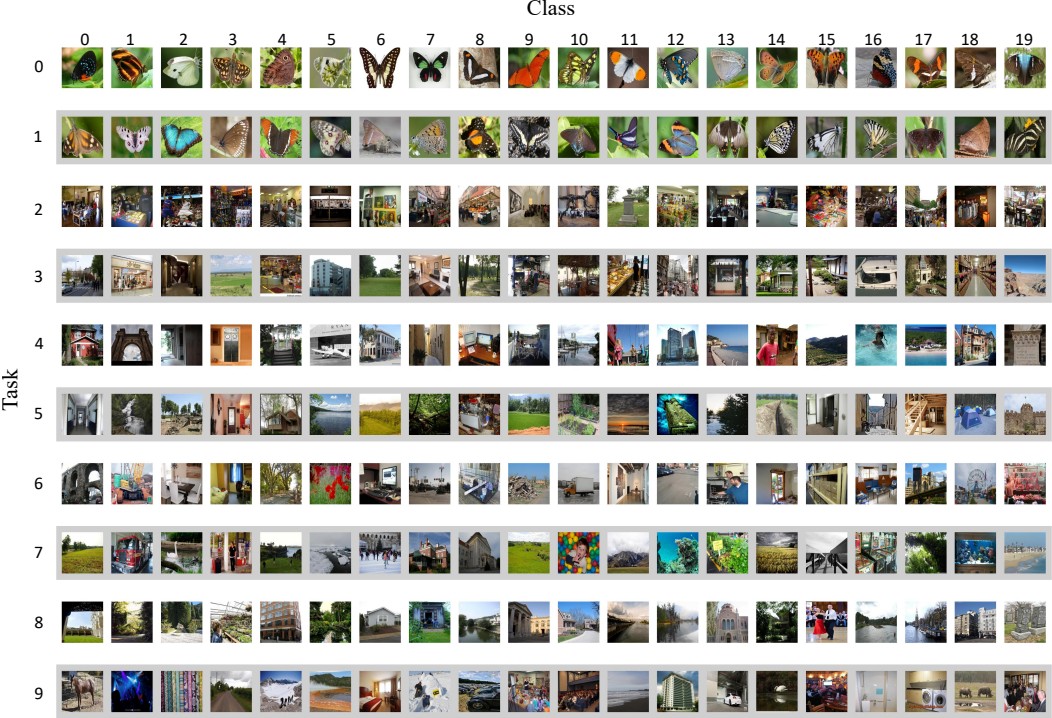

Figure 18: Sample images in a commonly challenging CLDyB sequence (seed-0).

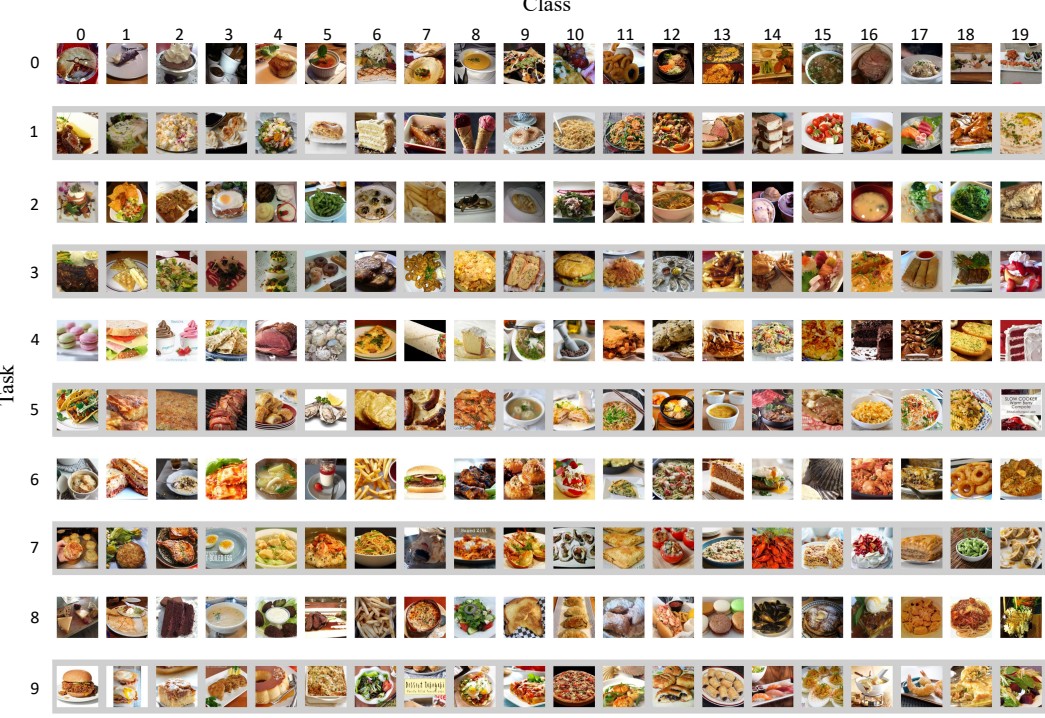

Figure 19: Sample images in a commonly challenging CLDyB sequence (seed-1).

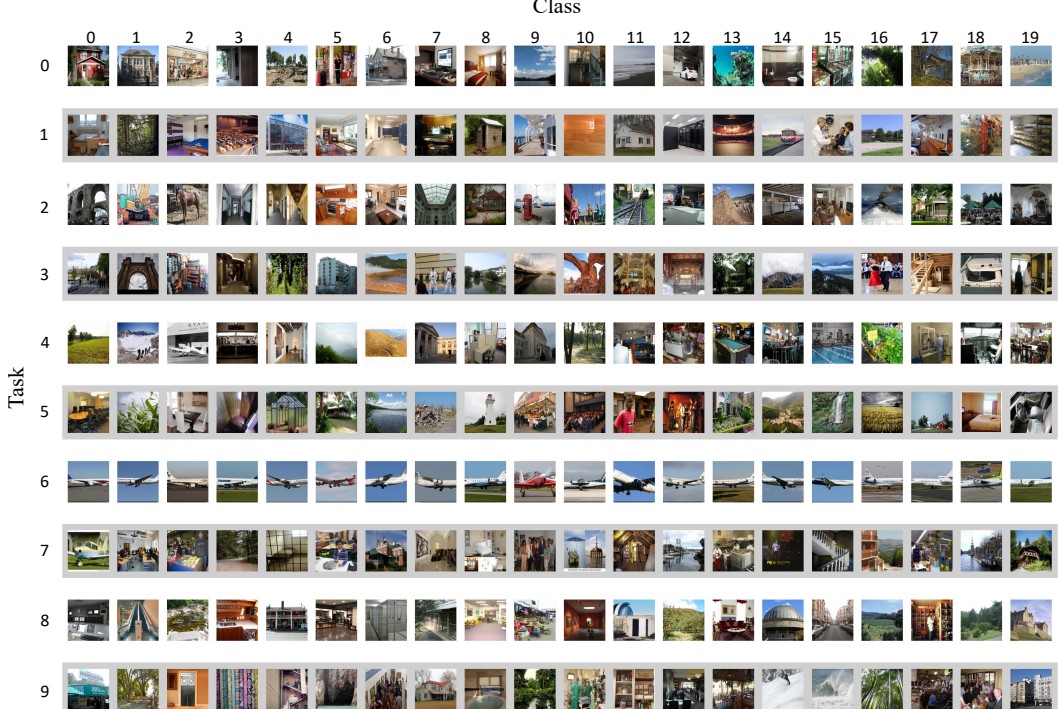

Figure 20: Sample images in a commonly challenging CLDyB sequence (seed-2).

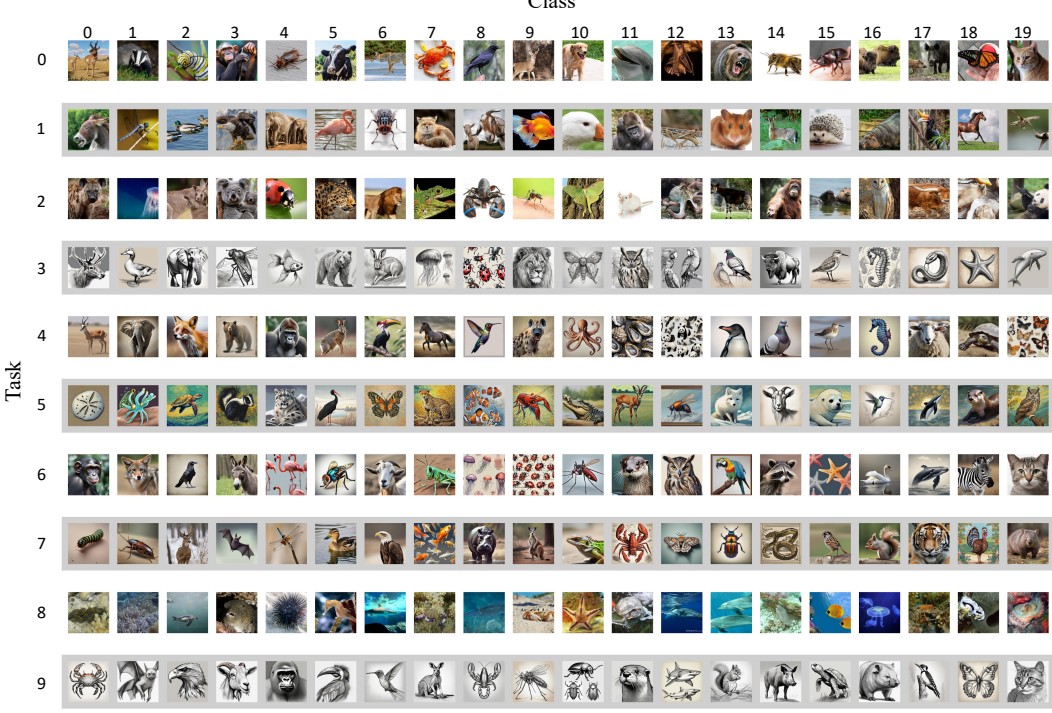

Figure 21: Sample images in a commonly challenging CLDyB sequence from the augmented data pool by AI-generated images after time step $t = 2$. Tasks consisting of AI-generated images are selected at time steps $\{3, 4, 5, 6, 7, 9\}$.

