# OpenReview forum: "CLDyB: Towards Dynamic Benchmarking for Continual Learning with Pre-trained Models"
_ICLR.cc/2025/Conference — ICLR 2025 Poster_

### Official Review · Reviewer_BrAw · 2024-11-01

**Soundness:** 3
**Presentation:** 3
**Contribution:** 3
**Rating:** 6
**Confidence:** 4

**Summary:**

The paper benchmarks different approaches in CL. The main contribution of this work is the use of optimization and tree search to come up with more challenging tasks.

**Strengths:**

Dynamic benchmarking seems like a novel idea.
The evaluation methodology is scalable
The metrics make sense
A well written paper
The graphics are very nice.

**Weaknesses:**

Why do we talk about benchmarking pre-trained models on CiFAR100, not even a CL dataset, nor an LLM dataset.

The training can only work with Pre-trained models, no starting from scratch.

The difficulty of choosing tasks is based on  some clustering approach, I would have thought that using some directional gradient to pick a task that pushes the cost in a particular direction to be much more reasonable for selecting the new tasks. .

**Questions:**

How does the approach ensure that the CL methods is not biased by the choice of the training sample. If uniform random is being used, I would understand it, however, in this case, there seems to be some form of search and it is easy to derail this search.

The metric is confusing, a smaller average accuracy seems to be the best choice, why? should it not be the other way round.

---

> ### Author Response · Authors · 2024-11-22
> **Thank you for your review**
>
> We would like to sincerely thank the reviewer for the time and effort invested in reviewing our manuscript. We believe that the comments and questions raised offer valuable insights that will significantly enhance the quality of our work.
>
> In response to the questions and concerns identified in the review, we have addressed each point in a separate comment section. Please let us know if our responses adequately address your concerns. We are more than happy to engage in further discussion.

---

> ### Author Response · Authors · 2024-11-22
> **Response to W1**
>
> > [W1] Why do we talk about benchmarking pre-trained models on CiFAR100, not even a CL dataset, nor an LLM dataset.
>
>
> Thank you for raising this question. We wish to respectfully point out the potential misunderstandings about using the CIFAR-100 dataset for CL training and evaluation:
> - **(Split) CIFAR-100**, first introduced in **[1]**, **is a widely used dataset for CL evaluation**.
> - For example, multiple prior works have evaluated their proposed CL methods on the so-called **Split-CIFAR-100 benchmark**, in which the CIFAR-100 dataset is randomly divided into a sequence of classification problems with non-overlapping class labels between the tasks **[2,3]**.
>
> **To avoid potential confusion** in the future, **we have revised the manuscript** and we now refer to the CIFAR-100 benchmark as the **Split-CIFAR-100** benchmark to indicate it is being used specifically in a continual learning setup.

---

> > ### Comment · Reviewer_BrAw · 2024-11-22
> > **Misleading answer**
> >
> > The computer vision community for the ease of use keeps using Cifar100 and. cifar10 dataset for CL problem.  While, this is true and you can provide 200 citations of this use, that does not mean, these are genuinely datasets appropriate for CL especially in the pre-trained model sense.
> >
> > Particularly because, both these datasets involve an assumption called independent and identically distributed, by the premise of CL, two tasks are internally linked otherwise, different orders of task sampling will give different results, which they actually do.
> > Moreover, you build you approach under the basic assumption as well,  where you use clustering and tree search to pick the next task. Without an inherent dependency, any sample should be enough and no need for your approach.
> >
> > Moreover, the paper talks about the pre-trained model scenario, in convolutional neural networks case, I can accept this, somewhat, in the pre-trained model case, i dont think, this is appropriate.
> >
> > So, this answer seems misleading.

---

> > > ### Author Response · Authors · 2024-11-24
> > > **Clarification on using CIFAR-100 as a baseline for comparison.**
> > >
> > > Thank you for your follow-up details regarding our previous response to [W1]. We are very pleased to see that we are in the same line with you, and your primary concern may stem from a potential misunderstanding **regarding the usage of CIFAR-100 in our paper**. We would like to humbly clarify that CIFAR 100 was **used solely as a baseline for comparison purposes**, to highlight the superiority of our proposed dynamic benchmark, CLDyB-seq (Table 1).
> > >
> > > - We selected CIFAR-100 as a baseline because, **as you have rightly pointed out**, it is on CIFAR-100 **`NOT` being an appropriate evaluation dataset for CL, particularly in the pre-trained model setup**. As stated in our Introduction:
> > >     - CIFAR-100 suffers from potential data contamination (Line 068);
> > >     - As a static benchmark, it is prone to benchmark overfitting (Line 079).
> > >
> > > - The comparison (Table 1) highlights that **our proposed benchmark indeed produces more reliable evaluation results than CIFAR-100**, as evidenced by the highest correlation with the Heldout evaluation.
> > > - Moreover, **all of our evaluation and analyses, in the experiment sections,** of different CL methods **are based exclusively on the results from our proposed benchmark**.
> > >
> > > We greatly appreciate your engagement and thoughtful feedback, and we hope this clarification addresses any remaining concerns. Please do not hesitate to reach out if further discussion is needed.

---

> ### Author Response · Authors · 2024-11-22
> **Response to W2**
>
> > [W2] The training can only work with Pre-trained models, no starting from scratch.
>
> Thank you for raising this interesting question regarding the compatibility of our CLDyB-pipeline with CL methods that train from scratch instead of using pre-trained models. **We wish to humbly emphasize** that:
> - **Our CLDyB-pipeline** is designed to be **versatile and compatible** for training and evaluating **any CL algorithms**, regardless of whether the CL algorithms choose to use pre-trained backbones.
> - **The choice to use a pre-trained model** as a starting point **is up to the CL algorithm itself**, and this decision **does not affect our benchmark's compatibility**. Essentially, training from scratch, with a randomly initialized backbone, is akin to using a backbone pre-trained on no data.
>
> **Our benchmark design primarily focuses on CL with pre-trained models** because:
> - Data contamination and benchmark overfitting **are less of an issue when considering training from scratch**.
> - As discussed in the Introduction, **pre-trained models**, due to their extensive training on large-scale datasets, **are more susceptible to performance saturation and overfitting** on standard, static CL benchmarks, which results in less reliable evaluation results of the CL methods and, in turn, **hinders genuine algorithmic innovation**.
> - **To address these shortcomings** of existing static benchmarks, we designed the CLDyB-pipeline to **dynamically generate CL task sequences to challenge increasingly powerful CL learners and foundation models as the field advances**, thereby providing a more rigorous evaluation of CL methods that leverage pre-trained models and facilitate rapid advancement in the field.
>
> In summary, while our benchmark is primarily designed for CL methods with pre-trained models, the CLDyB-pipeline's inherent versatility allows it to be effectively used for CL methods with a backbone trained from scratch.

---

> > ### Comment · Reviewer_BrAw · 2024-11-22
> > **W2**
> >
> > This is actually not a deal breaking weakness. So, OK.

---

> ### Author Response · Authors · 2024-11-22
> **Response to W3**
>
> > [W3] The difficulty of choosing tasks is based on some clustering approach, I would have thought that using some directional gradient to pick a task that pushes the cost in a particular direction to be much more reasonable for selecting the new tasks.
>
> Thank you for suggesting this plausible alternative approach for selecting difficult tasks. **We have considered this method** in the early stages of our project, specifically selecting the next (difficult) task based on the directional gradients that misalign the most with the existing tasks encountered in the sequences. **However**:
> - While this approach seems reasonable, we quickly realized that **it is not readily applicable to some CL methods that adopt a model-ensemble (parameter isolation) strategy**.
> - **For example**: DualPrompt [4] and PGP [5] both expand and optimize a new set of task-specific prompts in isolation for each incoming task.
> - Consequently, the gradient alignment between tasks, for these methods, no longer accurately reflect the difficulty of the tasks, since the **task gradients pertain to different sets of task-specific parameters that are not directly comparable**.
>
> That said, **to ensure flexibility and compatibility** of our benchmark and CLDyB pipeline, we instead proposed **a two-step approach** when constructing challenging CL task sequences, as we detailed in the paper:
> - **Step 1 sampling for difficult tasks**: We design two clustering approaches to construct intrinsically difficult tasks, **narrowing the task candidate space** for subsequent searches by eliminating redundancy.
> - **Step 2 search for difficult task sequences** : We utilize a tree search algorithm to select the next difficult task from the candidate pool, which is **driven by the reward defined in Equation 6**, in a trial-and-error manner.
>
> We note that our two-step approach **require access only to the outputs (predictions)** of the CL models to evaluate Equation 6 and compute the reward. Thus, our implementation treats the CL models being evaluated as **a black-box function**, denoted as $f^t =\mathcal{A}(f^{t−1} , \mathcal{T}^t)$ , as detailed in Line 13 of Algorithm 3 (Appendix). **This ensures high versatility** for our dynamic evaluation pipeline.

---

> > ### Comment · Reviewer_BrAw · 2024-11-22
> > **Misleading Answer**
> >
> > While this approach seems reasonable, we quickly realized that it is not readily applicable to some CL methods that adopt a model-ensemble (parameter isolation) strategy.
> > For example: DualPrompt [4] and PGP [5] both expand and optimize a new set of task-specific prompts in isolation for each incoming task.
> > Consequently, the gradient alignment between tasks, for these methods, no longer accurately reflect the difficulty of the tasks, since the task gradients pertain to different sets of task-specific parameters that are not directly comparable.
> >
> >
> > If this is true, there is no similarity between your tasks to actually cluster them. Especially if the task are being represented by two different parameter spaces, then, the clustering similarity measure does not exist. An assumption that seems to be directly violated by this answer.

---

> ### Author Response · Authors · 2024-11-22
> **Response to Q1**
>
> > [Q1] How does the approach ensure that the CL methods is not biased by the choice of the training sample. If uniform random is being used, I would understand it, however, in this case, there seems to be some form of search and it is easy to derail this search.
>
> Thank you for your question. Since our benchmarks are searched from a large collection of existing datasets, **most of these datasets have their official training, validation, and testing splits**. Therefore, we directly use these original training samples.
>
> **In cases where official splits are not available**, **we indeed employ uniform sampling** to create disjoint training, validation, and testing sets for these datasets

---

> > ### Comment · Reviewer_BrAw · 2024-11-22
> > **Unclear Answer.**
> >
> > The use of training, testing and validation has no link to the choice of different tasks at each new iterations. For instance, if you choose the second task, after already choosing the first task, there is an inherent bias in the tasks. This bias does not get resolved due to the choice of training testing and validation.
> >
> > If the first task is very difficult, the second task cannot go anywhere, because the model is already going to do poorly, this bias does not get resolved by choosing training testing and validation data.

---

> > > ### Author Response · Authors · 2024-11-24
> > > **Further clarification on inherent biases in CL**
> > >
> > > We sincerely thank the reviewer for further elaborating on this concern, which has helped us better understand your primary concern -- how to address bias from the searched task sequences CLDyB-seq, rather than focusing on how to choose training samples in each selected task as discussed in our previous response.
> > >
> > > - First and foremost, we fully acknowledge that the bias the reviewer is concerned about, i.e.,
> > >     >"...if you choose the second task, after already choosing the first task, there is an inherent bias in the tasks..."
> > >
> > >     is indeed urgent and widely recognized to account for **low plasticity** in CL [1]. Earlier tasks can indirectly affect subsequent tasks, as **a difficult task 1 may lead to a poor initialization for learning task 2**, which can hinder model performance on task 2.
> > >     - This bias persists even under uniformly random task sequences, as long as tasks vary in difficulty levels, as evidenced in [2].
> > >
> > > - Therefore, plasticity to combat such bias, potentially available in any task sequence, is a key evaluation criterion and **design objectives of CL models**, e.g.,  avoiding poor local optima from task 1 as the initialization for task 2.
> > >     - The **objective of a benchmark** is on the contrary to **even highlight such biases and reveal differences in how various CL models address them**. This is one of the key motivations for designing our dynamic benchmark, CLyDB, which is specifically intended to evaluate how different CL methods handle “difficult tasks” and thus the bias differently.
> > >
> > > - Different CL models, with varying levels of plasticity, mitigate the biases inherent in task sequences to differing degrees. Thus, it is not necessarily true that:
> > >     > "if the first task is very difficult, the second task cannot go anywhere",
> > >
> > >     and **the challenging sequences searched with respect to individual CL models by our CLDyB-pipeline are indeed different**.
> > >
> > >
> > >     - To illustrate this point, we conducted an additional experiment:
> > >         - We **first identified a very difficult task 1** by utilizing our proposed CLDyB-pipeline to construct a pool of candidate tasks: we trained and evaluated two different CL methods, CLSER and AFEC, starting from the pre-trained ViT-Base-Sup21K backbone on each candidate task 1, selecting the most difficult candidate as the task 1 based on the lowest accuracy achieved (0.785).
> > >         - Next, we investigated **how this inherent bias in this task 1 affects the accuracy of task 2** by training these two CL models **on a common set of candidate tasks** for task 2, which were again discovered through our CLDyB-pipeline.
> > >     - We compared the **task 2 performances of the two CL models**, as shown in the table below, where each row represents a different candidate task for task 2. Based on the results, we confirm that:
> > >         -   The **performance on task 2 can still vary significantly** for each CL model after training on the very difficult task 1. For example, AFEC achieved a performance range from **as high as 0.97 to as low as 0.627**.
> > >         -    The **same inherent bias** (training on the same, difficult task 1) **led to different relative performances** among the common set of task 2 candidates **for different CL models**. For instance, CLSER identified candidate 9 as the most difficult, while AFEC struggled most with candidate 4.
> > >     - By comparing the challening sequences found w.r.t. individual CL models, **we reveal the unique strengths and weaknesses of various CL models**, which can facilitate advancements in algorithmic design (please see **Figure 5** and experiment in **Line 428** for more detailed discussion).
> > >
> > > | Task 2 Candidate Index | Task 2 Acc AFEC | Task 2 Acc CLSER |
> > > |------------------------|------------------|------------------|
> > > | 1                      | 0.952            | 0.938            |
> > > | 2                      | 0.71             | 0.702            |
> > > | 3                      | 0.935            | 0.938            |
> > > | 4                      | **0.627** (Lowest) | 0.717            |
> > > | 5                      | 0.712            | 0.76             |
> > > | 6                      | 0.655            | 0.848            |
> > > | 7                      | 0.67             | 0.75             |
> > > | 8                      | 0.7              | 0.973            |
> > > | 9                      | 0.66             | **0.683** (Lowest) |
> > > | 10                     | **0.97** (Highest) | **0.978** (Highest) |
> > >
> > >
> > >
> > > [1] Riemer et al., Learning to Learn without Forgetting by Maximizing Transfer and Minimizing Interference, ICLR 2019
> > >
> > > [2], Dohare et al., Loss of plasticity in deep continual learning, Nature 632, 768–774 2024

---

> > > > ### Comment · Reviewer_BrAw · 2024-11-30
> > > > **Summary**
> > > >
> > > > From what I understand, with all the authors explanations, it appears that.
> > > >
> > > > 1. The paper provides a benchmark and the key contribution would be the software infrastructure within the pipeline
> > > > 2. Methodologically, the choice of the different is not novel but requires significant number of heuristics to get them working on each of the different applications and/datasets we use.
> > > > 3. Moreover, finding these harder tasks especially for pre-trained language models is going to be computationally prohibitive because, clustering all the data and figuring out difficulty of tasks for such huge datasets is virtually impossible. If it was possible, we would not need pre-trained models.
> > > >
> > > > if my understanding is correct, I think, I understand that this paper would be a good contribution in the sense of how to design more robust CL models. However, the claims in this paper are a little misleading especially in the sense of pre-trained models.
> > > >
> > > > With this in mind, I think the score I gave is appropriate, this is a good contribution.

---

> ### Author Response · Authors · 2024-11-22
> **Response to Q2**
>
> > [Q2] The metric is confusing, a smaller average accuracy seems to be the best choice, why? should it not be the other way round.
>
> Thank you for raising this concern.
>
> Recall that **our benchmark design aims to continuously and dynamically discover CL tasks and sequences that sufficiently challenge existing CL algorithms and pre-trained models** (lines 159, 239), as this approach:
> 1. **Alleviates data contamination** in CL evaluation with strong pre-trained models.
> 2. **Prevents benchmark overfitting** due to the dynamic nature of the constructed task sequences.
>
> Traditionally, **CL algorithms strive to maximize their performance**, which is reflected by higher metrics such as average accuracy. **To effectively challenge these CL algorithms**, we have adopted a strategy that **directly opposes this standard objective by searching for task sequences that minimize CL performance** (average accuracy).
>
> Consequently, **our CLDyB-seq selects candidate tasks** that result in the **lowest** average accuracy for the CL methods, as shown in Equation 4.
>
> **To prevent any potential confusion** regarding the interpretation of the metrics, we have also clarified in the experiment session that "Higher values (metrics) indicate better performance of the CL methods, yet it also implies that the benchmark poses less of a challenge for CL (**line 344**)".

---

> > ### Comment · Reviewer_BrAw · 2024-11-22
> > **Okay (Q2)**
> >
> > I think, this is clear.

---

> ### Author Response · Authors · 2024-11-22
> **References**
>
> [1] Rebuffi et al., iCaRL: Incremental Classifier and Representation Learning, CVPR 2017
>
> [2] Wang et al., Hierarchical decomposition of prompt-based continual learning: Rethinking obscured sub-optimality. NeurIPS, 2023
>
> [3] Zhang et al., Slca: Slow learner with classifier alignment for continual learning on a pre-trained model. ICCV 2023
>
> [4] Wang et al., Dualprompt: Complementary prompting for rehearsal-free continual learning, ECCV 2022
>
> [5] Qiao et al., Prompt Gradient Projection for Continual Learning, ICLR 2024

---

> ### Author Response · Authors · 2024-11-24
> **Clarification on functional task clustering**
>
> Thank you for your follow-up question.
>
> To begin, we respectfully point out that **in our Functional Task Clustering**, all candidate tasks are compared and **clustered in a shared representation space—defined by the same model**, $f^{t-1}_m$, with the **same parameters**— so **there is meaningful similarity** between these candidates for clustering. More specifically:
>
> Recall our **functional task clustering is primarily designed to cluster similar candidate tasks** and eliminate redundancy, thereby narrowing down the search space for subsequent tree search (MCTS) where more challenging tasks are selected. Note, this clustering process **does not directly select difficult tasks**. To reach our goal:
> - As shown in **Algorithm 2, Line 6**, we first **treat $f^{t-1}_m$ as a frozen, black-box function** for prediction on each candidate task $\mathcal{T}^t_i\in \mathbb{T}^t_{(g)}$, adhering to its original implementation for inference, to obtain the model's output distributions $p^{t-1}_m(y|x)$ for each sample in the candidate tasks.
> - We then average the model outputs over samples within the same candidate task **to compute the task's negative log-likelihood** as $-\frac{1}{|\mathcal{T}\_i^t|}\sum_{x,y\in\mathcal{T}\_i^t}\log p^{t-1}_m(y|x)$, which is **then subsequently used as the task representation for k-means clustering**.
> - It is crucial to note that the entire process **consistently uses the same $f^{t-1}_m$ across all candidates** and **does not involve training $f^{t-1}_m$ on specific candidate tasks**. Therefore, there is no model expansion for each different candidate task (for model-ensemble, parameter isolation CL methods). **In contrast**, the previously suggested approach would require task-specific training, i.e., $\mathcal{A}\_m(f_m^{t-1},\mathcal{T}\_i^t)\rightarrow f^{t}\_{m,i}\ \ \forall \mathcal{T}^t\_i\in \mathbb{T}^t\_{(g)}$, to obtain gradients relevant to different sets of task-specific parameters, i.e., $\nabla_{f^{t}\_{m,\boldsymbol{i}}},\ \ f^{t}\_{m,i}\neq f^{t}\_{m,j},\ \forall \mathcal{T}_i^t\neq \mathcal{T}_j^t$, which diminishes the reliability of gradient alignment as an indicator of task difficulty.
> - Consequently, **the same model $f^{t-1}_m$, and thus the same set of parameters, is employed to measure the similarity among different candidate tasks**, making this similarity meaningful for clustering.

---

> ### Author Response · Authors · 2024-12-04
> **Thank you for your positive feedback! & Our claims and contributions**
>
> Dear Reviewer BrAW,
>
> Thank you for your positive feedback and encouraging words! We appreciate your acknowledgment of our contributions in implementing the dynamic task selection pipeline, CLDyB-pipeline, and how it facilitates the design of more robust CL methods.
>
> In addition to the above, to guide a swift review of our contributions and claims, we offer a concise clarification below.
>
> **We would like to humbly highlight that our work is the first of its kind** in the following ways:
> - We **propose the idea of dynamic benchmarking** to address the issues of data contamination and unreliable evaluation results associated with standard CL benchmarks when **evaluating CL methods that utilize a pre-trained model as an advantageous starting point**.
> - We **provide a feasible implementation of a dynamic task selection pipeline**—in which tasks in sequences are dynamically selected to present sufficient challenges for the CL methods—to enable dynamic benchmarking. Regarding this implementation, we would also like to emphasize:
>     - The functional task clustering (Alg.2) requires **only forward passes** (i.e., no gradients) to compute the task negative log-likelihood of the M CL models. These log-likelihoods are then concatenated into an **M-dimensional vector** for each task, serving as features for k-means clustering. This approach makes the clustering **computationally efficient**.
>     - The greedy task sampling followed by functional task clustering (**Stage A** of the CLDyB pipeline) is designed to **drastically reduce the number of difficult candidate tasks** for subsequent tree search. As a numerical indication, Stage A can narrow down to 50 difficult tasks from the entire task space containing 6E^47 possible tasks (combination(2043,20)). Consequently, only this small portion of tasks is actually explored during the tree search. This design **ensures the pipeline's effectiveness** in searching for difficult tasks and task sequences (see Fig. 7 ablation) **while maintaining computational feasibility even for an enormous task space.**
>     - Our contributions and the pipeline are primarily focused on **providing a reliable evaluation** of CL methods that use pre-trained models in their algorithm design. Therefore, **the benefits associated with using pre-trained backbones as an advantageous starting point for CL are orthogonal to our contributions**.
>
>
> Moreover, **as results from applying our versatile CLDyB-pipeline** in two distinct scenarios:
> - We discover CL task sequences, **CLDyB-seq, that are commonly challenging** to a broad range of CL methods and empirically confirm that evaluation results made on our benchmark **readily translate to robust real-world performances**, **making these CLDyB-seq a suitable benchmark** for existing CL methods with pre-trained models.
> - We show that the CLDyB-pipeline can also be used **as a tool for analysis** by running it **on individual CL methods** to search for task sequences that are specifically challenging to the method. By analysing these individually challenging sequences, **CLDyB reveals the relative strengths and weaknesses of different CL methods**, facilitating rapid algorithmic development.
>
> We hope our work can make a meaningful first step towards developing more reliable evaluation protocols for CL methods in the era of pre-trained foundation models.
>
> Thank you so much for your invaluable insights and sustained engagement during the discussion period! We truly appreciate your time and expertise.

---

### Official Review · Reviewer_RbVT · 2024-11-04

**Soundness:** 2
**Presentation:** 3
**Contribution:** 3
**Rating:** 6
**Confidence:** 4

**Summary:**

This paper proposes a “dynamic” framework for evaluating continual learning algorithms in a way that addresses two issues: i) the potential data contamination inherent in the methodology of using pretrained models as a starting point for continual learning, and ii) lack of realism in prior benchmarks due to their static nature.

Specifically, the proposed benchmark is dynamic in that it forms sequences of tasks using M CL algorithms. These algorithms are used to determine 1) the most challenging tasks, and 2) the most challenging ordering of them. For the former, the authors use a Markov Random Field construction with pairwise potentials encoding the similarity between pairs of classes (using their prototypes in embedding space). They employ a greedy algorithm for picking K classes that are most similar (higher pairwise potentials), starting with a randomly-chosen class, to ensure enough diversity. They then perform additional clustering and sample uniformly from clusters, to ensure enough diversity in the set of K-way classification problems that are chosen. Then, for picking the hardest sequence, they formulate an online sequential decision-making problem where the reward is defined based on a notion of difficulty (increased forgetting and decreased task accuracy). They solve the problem using Monte-Carlo Tree Search (MCTS). To mitigate the expensive nature of MCTS (due to the rollouts, which in this case require training models on different sequences of tasks), they use a look-ahead variation that is more compute-efficient at the expense of an approximation bias.

They propose two modes of using their dynamic benchmark: the first finds task sequences that are “commonly challenging” for a pool of considered CL algorithms / backbones, while the second finds task sequences that are individually challenging for a given CL algorithm / backbone. They perform several analyses with different CL algorithms on different task sequences.

**Strengths:**

- The paper is for the most part well written and easy to follow (see some exceptions in Weaknesses and Questions below)

- The proposed benchmark is well motivated and well-thought out (though see some questions below regarding some exceptions) and seems like a useful contribution to the CL community.

- Empirically, the authors show that the proposed approach for finding “generally challenging” sequences leads to task sequences that are challenging for “held out” CL methods too (that weren’t used during the sequence selection), making it a meaningful benchmark for measuring progress of new algorithms.

- On the other hand, the sequences generated by the benchmark individually for different CL algorithms can lead to an understanding of that algorithm’s limitations, and capture similarities between different CL algorithms in terms of their failure modes, which is very interesting.

- Several interesting and thorough analyses are considered, including the ability to expand the benchmark using generated data, and ablation studies of various components.

**Weaknesses:**

- Some clarity issues. For example, under Equation 2, certain symbols (e.g. the set of algorithms and models A, and F) are defined that aren’t used in Equations 1 and 2, but become relevant later and it’s not clear how they are integrated into the quantities AFM and ALA – is there an average missing?

- Related to the above, in Section 4.1, the authors discuss the metrics used for evaluation and mention AA, AR and ALA, but not AFM. Why are there different metrics considered here now? Why is AFM excluded from evaluation?

- Another important clarity issue regarding the experimental setup: the authors mention CL algorithms and backbones, and for the former, discuss that they were selected to ensure high performance and sufficient diversity / coverage. But how were the backbone selected? Are all CL algorithms applied on the same pretrained backbone, or are there different pretrained models and architectures considered? This feels like a very important aspect that is not discussed.

- When defining intra-class difficulty, it’s not clear that the proposed operationalization of “difficulty” corresponds with the issue of data contamination, which the authors use as motivation for choosing difficult tasks. Instead, the chosen approach seems to correspond to capturing difficulty as the difficulty to discriminate between pairs of classes based on their embedding-space prototypes, which isn’t *directly* related to having seen those classes at pretraining time (it is to some extent of course since what is seen at pretraining time influences the embedding function). It would be great to add some discussion about this. An alternative approach would be to independently choose the K “hardest” classes, rather than picking based on pairwise relationships (this might be seen as more directly capturing potential data contamination). This could be done e.g. by picking the classes whose examples are predicted with the smallest possible confidence in a classification task containing all classes. Have the authors considered such alternative approaches?

- Regarding the results and experimental setup in Table 1, can’t one claim that “Heldout” isn’t truly held-out due to potential data contamination issues that the authors have pointed out elsewhere in the paper? Has care been taken to alleviate this concern here? If not, can these results regarding the correlation of performance between Heldout and other benchmarks be confounded by potential data contamination?

- In the caption of Figure 1, the authors claim that the figure shows the insufficient sensitivity to performance differences, but I don’t see how this argument can be made without knowing whether the evaluated methods do in fact perform differently. If we knew that, we could say it’s a limitation of the benchmark for not capturing this. But if we don’t know that, we can’t rule out that these methods are actually indistinguishable from one another in terms of their performance. Could you comment on this?

- Insufficient discussion / description of the chosen CL algorithms (and families of CL algorithms in general). It is hard to follow the discussion of e.g. the trade-offs of different algorithms (based on Fig 4) without any description of those algorithms. Consider swapping the order of section to have Related Work appear before the experiments to address this.

- Insufficient discussion of other benchmarks. While the authors describe some in the related work, they don’t discuss what blind spots of those benchmarks (that are also dynamic), the proposed benchmark addresses. Further, in analyses like the one in Table 1, it would have been great to compare against such dynamic benchmarks rather than static ones like CIFAR-100 and ImageNet-R.

- Related work: it would be great to also discuss Nevis’22 (see References below) which also is motivated by building a more realistic benchmark for continual learning.

- Relationship to related work: are the trade-offs observed in Figure 4 between different families of methods novel? Have prior work pointed out similar trade-offs? What are the new findings here?

- In Equation 4, why are these objectives chosen in particular, and why with that particular weighting?


Minor:

“While continue learning approaches” → “While continual learning approaches” (line 062)


References

Nevis’22: A Stream of 100 Tasks Sampled from 30 Years of Computer Vision Research. Bornschein et al. 2023.

**Questions:**

- The authors argue (at the start of Section 3.1) that when there is data contamination in the context of a pretrained model, the resulting high performance on CL tasks does not provide a meaningful comparison between CL algorithms. Why is this the case? I agree of course that if there is data contamination, this may lead to overall inflated performance (compared to had there not been data contamination), but why are the relative differences between the performance of different CL algorithms not meaningful?

- There seems to be a hypothesis that “hardest” task sequences are more realistic. But this isn’t always in the case: many practical sequences of interest may not be adversarially difficult. And it might not be the case that the relative ranking between CL methods at the “hardest” sequences is the same as that for average-case / realistically-occurring sequences. Is there some empirical evidence that hardest tasks align best with realistic scenarios (in terms of relative ranking of CL methods)? It may make sense to offer different levels of difficulty (e.g. easy, medium and hard), and observe how the relative rankings of CL methods vary across these?

- How compute intensive is it, overall, to find the hardest tasks and hardest sequence? Is the computational cost required a function of M? What are the relevant trade-offs to consider?

---

> ### Author Response · Authors · 2024-11-22
> **Thank you for your review**
>
> We would first like to express our genuine appreciation for the time and effort the reviewer has dedicated to providing such a detailed and in-depth review of our submission. We believe these comments and questions will contribute significantly to improving the overall quality of our work and guide us in highlighting its unique contributions more effectively.
>
> Regarding the weaknesses and questions in the review, we have responded to each in a separate comment section. Additionally, we have uploaded a revised manuscript that incorporates your feedback. Please let us know if our response has sufficiently addressed the concerns. We are more than happy to engage in further discussion.

---

> ### Author Response · Authors · 2024-11-22
> **Response to W1**
>
> > [W1] Some clarity issues. For example, under Equation 2, certain symbols (e.g. the set of algorithms and models A, and F) are defined that aren’t used in Equations 1 and 2, but become relevant later and it’s not clear how they are integrated into the quantities AFM and ALA – is there an average missing?
>
> Thank you for pointing out this potential clarity issue. In our original manuscript, we replaced the input argument $f^t$ to Equations 1&2 with the set of models $\mathbb{F}^t$ and refered to the resultant metrics as the mean AFA and mean ALA (line 140).
> To enhance clarity and resolve any ambiguities, **we have revised the manuscript with the following explanations** and mathematical notations around line 140:
> - The two metrics defined in Equation 1 and 2 are initially for a single CL model, $f^t$, i.e., $\mathtt{AFM}(\mathbb{T}^{<t+1},f^t)$ and $\mathtt{ALA}(\mathbb{T}^{<t+1},f^t)$.
> - We denote the average AFM and ALA over CL $M$ models as $\mathtt{AFM}(\mathbb{T}^{<t+1},\mathbb{F}^t)=\frac{1}{M}\sum^{M}\_{m=1}\mathtt{AFM}(\mathbb{T}^{<t+1},f^t\_m)$ and $\mathtt{ALA}(\mathbb{T}^{<t+1},\mathbb{F}^t)=\frac{1}{M}\sum^{M}\_{m=1}\mathtt{ALA}(\mathbb{T}^{<t+1},f^t\_m)$, respectively -- replacing the input argument $f^t$ to Equations 1&2 with the set of models $\mathbb{F}^t=\\{f^t_m\\}\_{m=1}^{M}$.
> -  These averaged notations are subsequently used in the text.

---

> ### Author Response · Authors · 2024-11-22
> **Response to W2**
>
> > [W2] Related to the above, in Section 4.1, the authors discuss the metrics used for evaluation and mention AA, AR and ALA, but not AFM. Why are there different metrics considered here now? Why is AFM excluded from evaluation?
>
> We apologize for the confusion caused. **Essentially, AR and AFM are the same evaluation metric** -- both reflect forgetting in the CL model -- with the relationship **AR = -AFA** as stated in footnote 1, line 376.
>
> To avoid confusion in the future, **we have updated the footnote** with an explicit explanation on why this negation was done for AFM:
> - (**A**verage **R**etenttion) is equivalent to negative **A**verage **F**orgetting **M**easure; The negation is applied to ensure that a higher value of the metric indicates better performance of the CL method, aligning with the other two evaluation metrics, AFA and ALA.

---

> ### Author Response · Authors · 2024-11-22
> **Response to W3**
>
> > [W3] Another important clarity issue regarding the experimental setup: the authors mention CL algorithms and backbones, and for the former, discuss that they were selected to ensure high performance and sufficient diversity / coverage. But how were the backbone selected? Are all CL algorithms applied on the same pretrained backbone, or are there different pretrained models and architectures considered? This feels like a very important aspect that is not discussed.
>
> We primarily conducted experiments using the ViT-Base-Sup21K pre-trained on ImageNet-21k in searching for CLDyB-seq. This is because,
> - as an initial release of the commonly challenging CLDyB-seq as a fixed benchmark dataset, we would like to first address the needs of most researchers in the CL community by **focusing on the most popular pre-trained backbone used in recent works**, which is the the ViT-Base-Sup21K **[1,2]**;
> - in addition, to ensure a fair comparison between different CL methods at evaluation, we also maintained using the same pre-trained backbone across all CL methods.
>
> Nevertheless, it is indeed possible to apply the CLDyB-pipeline to target CL methods with alternative pre-trained backbones other than the ViT-Base-Sup21K, thanks to the high versatillity of our dynamic task selection CLDyB-pipeline -- a major contribution of our work — which is generally applicable to all CL algorithms and pre-trained backbones. By **incorporating different levels of diversity in the pre-trained backbones** (in conjunction with various CL methods) when running the CLDyB-pipeline, we can obtain commonly challenging task sequences at different granularities.
>
> - Switching to the **CLIP ViT-Base Backbone**
>     - In our original submission, we have evaluated the CLDyB-pipeline using the CLIP ViT-Base vision backbone, which was pre-trained on a significantly larger, web-scraped dataset, namely YFCC100M, instead of ImageNet-21k.
>     - The results, presented in **Appendix, Figure 11**, demonstrate that the CLDyB-pipeline is still capable of identifying commonly challenging CL sequences, as evidenced by performance drops of -10%, -8%, and -2% in average final accuracy, average retention, and average learning accuracy, respectively, of the CL methods.
>
> - Switching to **a Mixture of the CLIP ViT-Base and ViT-Base-Sup21K Backbones**
>     - During the response period, we conducted further experiments by incorporating a combinatorial mixture of different pre-trained backbones and CL methods and apply the CLDyB-pipeline to this resultant mixture in search for commonly challening CLDyB-seq.
>     - The results, presented in **Appendix, Figure 12**, demonstrate that the CLDyB-pipeline is still capable of identifying challening CL sequences, as evidenced by performance drops of -10%, -4%, and -7% in AFA, AR, and ALA, respectively, of the CL methods.
>     - Furthermore, the evaluation results of individual CL methods, as shown in **Appendix, Figure 13**, demonstrate that CLDyB-seq effectively challenges each CL method and successfully generalizes to unseen combinations of CL methods and pre-trained backbones.
>
> While our initial plan is to open-source CLDyB-seq as a fixed benchmark of task sequences designed to challenge current state-of-the-art CL methods using the ViT-Base-Sup21K and CLIP-ViT-Base backbones, we may consider releasing multiple tracks of commonly challenging CL sequences in future releases. Each track would target a family of pre-trained backbones that are popular in the CL community.

---

> ### Author Response · Authors · 2024-11-22
> **Response to W4**
>
> > [W4] When defining intra-class difficulty, it’s not clear that the proposed operationalization of “difficulty” corresponds with the issue of data contamination, which the authors use as motivation for choosing difficult tasks. Instead, the chosen approach seems to correspond to capturing difficulty as the difficulty to discriminate between pairs of classes based on their embedding-space prototypes, which isn’t directly related to having seen those classes at pretraining time (it is to some extent of course since what is seen at pretraining time influences the embedding function). It would be great to add some discussion about this. An alternative approach would be to independently choose the K “hardest” classes, rather than picking based on pairwise relationships (this might be seen as more directly capturing potential data contamination). This could be done e.g. by picking the classes whose examples are predicted with the smallest possible confidence in a classification task containing all classes. Have the authors considered such alternative approaches?
>
> **Data contamination is indeed challenging to measure directly**, as it involves understanding the overlap between pre-training data and downstream tasks while pre-training data is oftentimes inaccessible. Therefore, **heuristic scores**, such as the suggested approach of picking low-confidence classes, and our approach of evaluating class separability in the embedding space, **are necessary for an approximation**.
>
> **Regarding the proposed alternative**
>
> - We appreciate the alternative approach suggested by the reviewer. **Predictive confidence** has indeed been shown to be **closely related to the issue of data contamination** [3,4].
> - However, we respectfully **note that the suggested alternative** of selecting low-confidence classes based on model's predictive distribution **is highly correlated with our method** of identifying difficult tasks based on class pairwise separability within that task, **as a class's pairwise separability with other classes in the embedding space directly affects its predictive distribution, hence confidence**.
>     - **Intuitively**, the predictive confidence of a class in a classification problem is heavily influenced by its most confusing classes, as **those classes that are clearly separated will contribute minimally to its softmax predictive distribution**.
>     - **As a thought experiment**, consider a total of three classes where class 1 is perfectly separated from classes 2 and 3 in the embedding space.
>         - In a 3-way classification, the average predictive distribution for samples in [class 1,class 2, class 3], could be [[1,0,0], [0,0.3,0.7],[0,0.4,0.6]], as samples in class 1 will be perfect predicted.
>         - The prediction confidence  for each class is [1, 0.3 ,0.6], and the entropy for each class is [0, 0.882, 0.971].
>         - To select two classes to construct a 2-way classification problem, our method based on embeddings separability would pick class 2 and class 3; while the suggest alternative would also pick class 2 and class 3 due to their lower prediction confidence (or higher entropy)
>      - During the discussion period, **we conducted an additional experiment to verify the similarity between the two approaches**.
>        - Given a pre-trained backbone, we employed both our Greedy Task Sampling (Algorithm 2, Appendix) and the suggested approach to pick out a 50-way classification task from the 2043 available classes.
>        - We recorded **the number of overlaps between the two sets of 50 classes picked by the two approaches**, which were **41** and **43** for the ViT-Base-Sup21K and the CLIP-ViT-Base pre-trained backbones, respectively.
>        - The **large intersection** indicate that the classes selected by our approach highly coincide with those having low predictive confidence in an all-class classification problem.
>
> - Therefore, we conclude that the two approaches are highly related and both reflect the issue of data contamination through the model's predictive distribution.

---

> ### Author Response · Authors · 2024-11-22
> **Response to W5 and Q1**
>
> > [W5] Regarding the results and experimental setup in Table 1, can’t one claim that “Heldout” isn’t truly held-out due to potential data contamination issues that the authors have pointed out elsewhere in the paper? Has care been taken to alleviate this concern here? If not, can these results regarding the correlation of performance between Heldout and other benchmarks be confounded by potential data contamination?
>
> >[Q1] The authors argue (at the start of Section 3.1) that when there is data contamination in the context of a pretrained model, the resulting high performance on CL tasks does not provide a meaningful comparison between CL algorithms. Why is this the case? I agree of course that if there is data contamination, this may lead to overall inflated performance (compared to had there not been data contamination), but why are the relative differences between the performance of different CL algorithms not meaningful?
>
> We appreciate these thoughtful comments regarding potential data contamination issues in the Heldout task sequences in Table 1, as well as in existing static benchmarks in general.
>
> **Regarding [W5] Data Contamination the in Heldout, Table 1**
> - To begin, we would like to clarify that **care has indeed been taken** to avoid data contamination in the Heldout to the best of our ability. The Heldout task sequences are constructed from datasets that have never appeared in either the CLDyB-sequences or the pre-trained dataset of the vision backbones. This ensures that, **at a dataset level, the pre-training, CLDyB-sequences, and Heldout datasets are disjoint**.
> - While we agree with the reviewer in acknowledging that the issue of data contamination is difficult to avoid completely in practice, we humbly emphasize that the **central message** we wish to convey through the results **in Table 1** is that the **CLDyB-seq produces reliable evaluation results**, which are a better indication of the relative performance of CL methods in an unseen real-world CL scenario (line 392). That said, we believe that **even if there is some unavoidable contamination in the Heldout, it does not undermine delivery of our central message because**:
>     - **There could only be minor contamination** in the Heldout due to its datasets being disjoint with others; and as evident by the numerical results in Table 1, the performances on Heldout are still **substantially lower** in comparison to those on CIFAR-100.
>     - When there is such minor data contamination, **we agree with the reviewer** in that although this could lead to slightly higher performance for all methods, **the relative performance (or relative ranking), however, remain quite consistent, and therefore is still meaningful**.
>     - We also acknowledge that in a realistic downstream CL application (i.e., CL model deployment time), we cannot completely rule out the possibility that some tasks can be similar to the ones in the pre-training dataset (being slightly contaminated).
>
> **Regarding [Q1] inflated performances being less meaningful, Section 3.1**
> - As mentioned above, **the real problem arises**, making relative performances less meaningful, **when data contamination is no longer minor** but becomes a dominating factor affecting the evaluation results (when most downstream tasks exhibit significant similarity with the pre-trained data).
> - In this scenario, performances are inflated and dominated by data contamination: methods can easily improve their performance by focusing **solely on exploiting the pre-trained models rather than designing generalizable algorithmic techniques**, all achieving very high performance on the evaluation benchmark [5].
> - Therefore, **there is an issue of performance insensitivity**, which prevents us from observing the full picture and conducting a rigorous and fair comparison between the CL methods.
> - **We have rephrased the sentence in Section 3.1 accordingly** for a more accurate expression of our argument. Thank you again for your feedback!

---

> ### Author Response · Authors · 2024-11-22
> **Response to W6**
>
> > [W6] In the caption of Figure 1, the authors claim that the figure shows the insufficient sensitivity to performance differences, but I don’t see how this argument can be made without knowing whether the evaluated methods do in fact perform differently. If we knew that, we could say it’s a limitation of the benchmark for not capturing this. But if we don’t know that, we can’t rule out that these methods are actually indistinguishable from one another in terms of their performance. Could you comment on this?
>
> Thank you for your thoughtful comment on whether insufficient sensitivity to performance differences on existing benchmark in Figure 1 is due to CL methods being actually indistinguishable from one another in terms of their performance.
>
> While the CL methods in Figure 1 (and also Table 1) show very similar performance on standard CL benchmarks, **our empirical results in the paper reveals that these methods can actually behave quite differently** under more challenging CL scenarios, such as the CLDyB-seqs and the Heldout datasets. **Specifically, with numerical results shown in Table 1**:
>
> - For groups like \{RanPAC, HidePrompt\} and \{DualPrompt, PGP, LAE\}, while the methods within each group performed similarly on the CIFAR-100 benchmark, their **performance gap widened significantly on the CLDyB-seqs**. For instance, the performance difference between RanPAC and HidePrompt **increased from 0.4% to 5.6%**.
> - The **relative ranking** of CL methods on CIFAR-100 and ImageNet-R **changed significantly** when evaluated on the CLDyB-seqs and the Heldout benchmark. Most notably, ER, which performed worst on CIFAR-100 and ImageNet, emerged as one of the top-performing methods on the CLDyB-seq and the Heldout, demonstrating strong robustness under more demanding CL scenarios.
>
> These **observable differences in evaluation results** reveal limitation in existing benchmark for only capturing part of the overall picture, failing to reflect the variations in performance under more challenging and realistic CL settings.

---

> ### Author Response · Authors · 2024-11-22
> **Response to W7**
>
> > [W7] Insufficient discussion / description of the chosen CL algorithms (and families of CL algorithms in general). It is hard to follow the discussion of e.g. the trade-offs of different algorithms (based on Fig 4) without any description of those algorithms. Consider swapping the order of section to have Related Work appear before the experiments to address this.
>
> Thank you for this great suggestion, which we believe will make our manuscript much more reader-friendly.
>
>  **We have updated the manuscript with the following changes**:
> - We have **positioned Related Work before Experiments**, as recommended.
> - In the original manuscript, due to space constraints, we deferred a more detailed discussion of conventional CL algorithms to **Appendix B.1**. Additionally, the CL methods evaluated in our experiments were detailed in **Appendix B.2**. We have now **explicitly highlighted these in the Related Work** section for earlier reference.

---

> ### Author Response · Authors · 2024-11-22
> **Response to W8**
>
> >[W8] Insufficient discussion of other benchmarks. While the authors describe some in the related work, they don’t discuss what blind spots of those benchmarks (that are also dynamic), the proposed benchmark addresses. Further, in analyses like the one in Table 1, it would have been great to compare against such dynamic benchmarks rather than static ones like CIFAR-100 and ImageNet-R.
>
> We first provide **a more detailed discussion on the limitations of the dynamic benchmarks mentioned in the related work section** below:
>
> - **Requiring crowd-sourced effort for dynamic evaluation**: Dynabench [6] and DynaBoard [7] are two early dynamic benchmarks that create evolving evaluation samples for LLMs in NLP tasks by utilizing **human-in-the-loop** for generating adversarial testing questions that fool the language models. In contrast, CLDyB-seq relies on a versatile CLDyB-pipeline for fully automatic dynamic CL tasks generation.
> - **Targeting specific NLP question domains which are not readily applicable to CL problems**: While Dyval [8] and Benchmark Self-Evolving [9] overcome the issue of crowd-sourced data collection by respectively utilizing directed acyclic graphs and a multi-agent framework for dynamic test set generation, it remains unclear how the two dynamic data generation techniques can be applied in the CL settings:
>   - The directed acyclic graphs in Dyval can **only target reasoning tasks in natural language for LLMs**, e.g., maths, logical reasoning (True or False), as the nodes in the graphs need to correspond to logic units whose relationships are expressed through edge connections. **However**, there is no such obvious logical relationship in CL tasks and sequences that can be modelled through direct acyclic graphs.
>   - On the other hand, the multi-agent framework in Benchmark Self-Evolving **is built on 4 GPT-4 LLMs agents** which are designed to dynamically filter and generate new testing questions from existing test sets through interactions. **Unfortunately**, such GPT-4 agents are yet incapable of generating sufficiently challenging CL task sequences w.r.t. a given set of state-of-the-arts CL methods and pre-trained models.
>
> Therefore, we conclude that **although these benchmarks are dynamic, none of them are readily applicable** and address the challenges in the setting of class-incremental continual learning. We are thus highly motivated to design **the first-ever dynamic benchmark for CL** in order to address this research gap.

---

> ### Author Response · Authors · 2024-11-22
> **Response to W9**
>
> >[W9] Related work: it would be great to also discuss Nevis’22 (see References below) which also is motivated by building a more realistic benchmark for continual learning.
>
> Thank you for pointing out the relevance of Nevis'22 in the context of building more realistic benchmarks for continual learning. We appreciate the opportunity to discuss this work in relation to our proposed CLDyB framework. We give a detailed comparion between the two **from three key aspects** below
>
> - **Benchmark Design Principle**:
>     - Nevis'22 presents **a historical stream** of 100 tasks manually constructed from 30 years of computer vision research, aiming **to provide a comprehensive evaluation** of continual learning methods.
>     - In contrast, the CLDyB-seq are dynamically and **automatically generated**, aiming **to present sufficient challenges** for even the most powerful CL algorithms and pre-trained backbones at the current time.
> - **Static vs. Dynamic Benchmark**:
>     - Nevis'22 is a **static** benchmark constructed through manual data selection, which **could eventually suffer from benchmark over-fitting** due to the advancement of either CL algorithms or pre-trained model.
>     - On the other hand, CLDyB-seq is **dynamic** and can be periodically updated to **tailor to the evolving states of state-of-the-art CL methods and pre-trained models**. This is made possible by our expandable CLDyB-pool and dynamic task selection CLDyB-pipeline.
> - **Versatility beyond the benchmark dataset itself**：
>     - In addition to releasing the CLDyB-seq as benchmark datasets. **Our work offers a general dynamic benchmark framework**, namely the CLDyB-pipeline, and demonstrates its versatility through **two important use cases**. Specifically:
>       - Finding **commonly challenging** CLDyB-seq, which **enhances** the likelihood of **algorithmic developments** made with our benchmark translating into strong real-world performance.
>       - Finding **individually challenging** CLDyB-seq, as a tool for analysis, for comparing and better **understanding the strengths and weaknesses of specific CL methods**, thereby facilitating rapid advancement of the field.
>
> In summary, while both Nevis'22 and CLDyB aim to improve the realism of continual learning benchmarks, our approach offers a unique contribution by focusing on dynamic task generation and addressing specific challenges related to pre-trained models in CL. We believe that these complementary efforts will collectively advance the field of continual learning.

---

> ### Author Response · Authors · 2024-11-22
> **Response to W10**
>
> >[W10] Relationship to related work: are the trade-offs observed in Figure 4 between different families of methods novel? Have prior work pointed out similar trade-offs? What are the new findings here?
>
> We would like to first humbly emphasize that **we are among the first works to offer a comprehensive comparison** between state-of-the-art CL methods specifically designed to utilize pre-trained models (PTMs). To the best of our knowledge, the **only other work** to carry out a similar comprehensive evaluation is a recent survey [2].
>
> While **the survey** compared the empirical performance of CL methods **mainly in terms of average accuracy**, our **Figure 4** provides the comparison **from multiple different aspects**, based on the results obtained on the CLDyB-seq. Figure 4 thus offers deeper insights into the unique (and novel) trade-offs and findings of these CL methods. **We highlight below the findings that are not presented in the survey** for better illustration:
>
> 1. **On Sophistication vs. Robustness**: Figure 4 points out that more sophisticated methods tend to improve performance over their predecessors but are not necessarily more robust. For example, PGP shows improvements over DualPrompt, but this does not translate to increased robustness. This observation adds a new dimension to the understanding of the trade-offs in CL methods.
> 2. **On Model Plasticity or Forward Transfer**: Exemplar-replay methods like ER and CLSER display significantly higher model plasticity compared to others. In contrast, parameter-efficient fine-tuning (PEFT) methods such as HidePrompt and LAE exhibit limited ALA, indicating poorer forward transfer ability.
> 3. **On Memory Efficiency**: Our results note that variants of replay methods, such as HidePrompt and RanPAC, although they generally demonstrate greater resistance to forgetting, are, however, markedly less memory efficient, which is a trade-off that is further emphasized here
> 4. **On Competitiveness of Simple Experience Replay**: Simple experience replay methods, such as ER, although they do not achieve the highest performances in terms of final AA and ALA, are recognized as the top overall CL method when evaluated on our challenging CLDyB-seq. This finding highlights the adaptability of simple experience replay methods across various dimensions, and encourages the CL community to focus on multiple aspects of algorithm design.
>
> **These findings collectively contribute to a more nuanced understanding of the trade-offs involved in using different CL methods with PTMs**.

---

> ### Author Response · Authors · 2024-11-22
> **Response to W11**
>
> >[W11] In Equation 4, why are these objectives chosen in particular, and why with that particular weighting?
>
> We would like to emphasize that **Equation 4 encapsulates our primary goal** in designing the CLDyB-pipeline: to "dynamically construct continual learning task sequences **that pose significant challenges** to M CL models across all time steps (line 240)".
>
> Traditionally, **the two main learning objectives** of a CL algorithm are to **minimize forgetting** and **maximize transfer**, which are quantified by the Average Forgetting Measure (**AFM**) and Average Learning Accuracy (**ALA**) metrics, respectively. **These metrics are standard evaluation tools** commonly used in prior works [10,11].
>
> **To effectively challenge the CL algorithms**, we have chosen a direct approach by searching for task sequences that minimize CL performance **by directly opposing these two standard objectives**.
>
> Without making explicit assumptions, **we treat forgetting and transfer as equally important objectives for CL**. Therefore, we adopted an unweighted summation in Equation 4. However, the search objective can be adjusted to a weighted sum of the two if one wants to prioritize one objective over the other.

---

> ### Author Response · Authors · 2024-11-22
> **Response to Q2**
>
> > [Q2] There seems to be a hypothesis that “hardest” task sequences are more realistic. But this isn’t always in the case: many practical sequences of interest may not be adversarially difficult. And it might not be the case that the relative ranking between CL methods at the “hardest” sequences is the same as that for average-case / realistically-occurring sequences. Is there some empirical evidence that hardest tasks align best with realistic scenarios (in terms of relative ranking of CL methods)? It may make sense to offer different levels of difficulty (e.g. easy, medium and hard), and observe how the relative rankings of CL methods vary across these?
>
> **Regarding the hypothesis that “hardest” task sequences are more realistic**:
>
> - Thank you for raising this important question. **To clarify, our hypothesis is** that evaluation results and algorithmic developments made on our challenging ('hardest') CLDyB-seq benchmark **are more likely to translate into strong real-world performance**. However, **this does not directly imply** that "hardest task sequences are more realistic" or that they mirror real-world CL sequences.
> - Our hypothesis is motivated by the intuition that **it is generally easier to generalize from hard to easy problems** rather than the reverse. The design principle of our benchmark follows this intuition, suggesting that evaluation results on our **"all hard tasks"** CLDyB-seq can be interpreted as **a worst-case performance** of a CL method -- a performance guarantee.
> - In contrast, evaluation results and algorithmic developments made on standard, easier benchmarks may not offer such a worst-case performance guarantee. They could perform poorly on challenging tasks, regardless of their success on easier ones.
> - The worst-case performance guarantee is particularly appealing in real-world applications that demand reliability. **By continuously improving this worst-case performance, we can ensure genuine algorithmic advancement.**
>
> **Regarding offering various difficulty levels in CLDyB-seqs:**
>
> - Thank you for this great suggestion; we believe it is an excellent idea!
> - **To incorporate varying difficulty levels** into the CLDyB-pipeline when searching for task sequences, **we propose the following implementation**:
>     - Instead of selecting the candidate task with the highest reward at each time step (greedy selection), as described in Equation 6, **we adopt a probabilistic approach**. The sampling probability of choosing a particular task is made **proportional to the task's reward**.
>     - To introduce varying difficulty levels, we employ an additional **temperature scaling factor** in the sampling process **as a hyperparameter to control the bias towards selecting the most challenging task**.
>     - A temperature of 0 corresponds to a greedy selection, while a higher temperature makes all candidate tasks more equally likely to be selected.
>     - All other components of the CLDyB-pipeline remain unmodified.
>
> - Based on this approach, we created two additional CLDyB-seqs: 'medium' and 'easy', with the original CLDyB-seq serving as the 'hard' version for evaluation. **The results are presented in the table below**:
> - |  Accuracy (rank)            | Hard   | Medium | Easy  | Heldout |
>   |--------------|--------|--------|-------|--------|
>   | RanPAC       | 56.9 (3)  | 62.2 (2)  | 77.0 (3)  | 81.0 (3)  |
>   | HidePrompt   | 62.5 (1)  | 67.6 (1)  | 80.1 (1)  | 84.9 (1)  |
>   | DualPrompt   | 41.9 (9)  | 53.7 (6)  | 69.0 (9)  | 70.8 (8)  |
>   | PGP          | 44.0 (8)  | 52.0 (7)  | 69.2 (8)  | 68.7 (9)  |
>   | LAE          | 48.1 (7)  | 50.9 (9)  | 70.1 (7)  | 71.1 (7)  |
>   | SLCA         | 56.2 (4)  | 59.7 (5)  | 77.2 (2)  | 80.3 (4)  |
>   | ER           | 54.8 (5)  | 60.5 (4)  | 76.3 (4)  | 79.9 (5)  |
>   | CLSER        | 57.0 (2)  | 60.7 (3)  | 75.4 (5)  | 81.1 (2)  |
>   | AFEC         | 49.5 (6)  | 51.9 (8)  | 72.5 (6)  | 76.3 (6)  |
>   | Spearman's rho | 0.983     | 0.833    | 0.867  | n/a    |
>   | Kendall's tau          | 0.944     | 0.667    | 0.722  | n/a    |
>     - As expected, the performance of **all methods improved as the difficulty decreased**.
>     - Performance on the **hardest CLDyB-seq indeed serve as a lower bound** for each CL method.
>     - Although there is some minor shuffling in the relative rankings,  the evaluation results for all three versions of CLDyB-seq still exhibit much **higher correlation with the Heldout** data compared to the standard benchmarks shown in Table 1.
>
> Once again, we vey much appreciate this awesome suggestion. We have included this discussion and results with varying difficulty levels in **Appendix D.1** (due to the current page limit), in the revised manuscript.

---

> ### Author Response · Authors · 2024-11-22
> **Response to Q3**
>
> > [Q3] How compute intensive is it, overall, to find the hardest tasks and hardest sequence? Is the computational cost required a function of M? What are the relevant trade-offs to consider?
>
> **In terms of the total compute memory cost**:
> - Running the CLDyB-pipeline on M learners is approximately equivalent to training and evaluating these M CL models on a fixed dataset under standard CL settings.
> - This is because standard CL training and evaluation are only carried out during the tree search, and both the greedy task sampling and functional clustering **primarily involve forward passes only, which introduce negligible memory overhead**.
> - Moreover, **the CLDyB-pipeline does not need to be run frequently**; it only needs to be executed periodically to accommodate the latest advancements in CL algorithms and pre-trained backbones. This periodic execution **makes the computational cost quite manageable over the long term**.
>
> **Regarding a relevant trade-offs on M**:
> - The **diversity** of the M CL learners, **controls the trade-off between granularity and generalization of the challengeness in the discovered CLDyB-seq**, as the challengeness of the resultant CLDyB-seq is dependent on the CL learner given for evaluation. Specifically:
>     - A larger M will not necessarily result in a more difficult sequence for a specific CL learner compared to the individual CLDyB-seq discovered using that particular learner alone when M=1, as the sequences are specifically tailored to that learner.
>     - However, the difficulty is likely to generalize better to some unseen learners in M due to boarder coverage of diverse CL techniques in larger M.

---

> ### Author Response · Authors · 2024-11-22
> **Response to Minor**
>
> > [M1] “While continue learning approaches” → “While continual learning approaches” (line 062)
> - Thank you for your attention to details, we have changed 'continue' $\rightarrow$ 'continual' in the Introduction to ensure consistency.

---

> ### Author Response · Authors · 2024-11-22
> **References**
>
> [1] Wang et al., Dualprompt: Complementary prompting for rehearsal-free continual learning, ECCV 2022
>
> [2] Zhou et al., Continual Learning with Pre-Trained Models: A Survey, IJCAI 2024
>
> [3] Xu et al., Benchmarking Benchmark Leakage in Large Language Models
>
> [4] Yeom et al., Privacy Risk in Machine Learning: Analyzing the Connection to Overfitting, IEEE CSF 2018
>
> [5] Galashov et al., Continually learning representations at scale, CoLLAs 2023
>
> [6] Kiela et al., Dynabench: Rethinking Benchmarking in NLP. NAACL 2021
>
> [7] Ma et al., Dynaboard: An Evaluation-As-A-Service Platform for Holistic Next-Generation Benchmarking, NeurIPS 2021
>
> [8] Zhu et al., DyVal: Dynamic Evaluation of Large Language Models for Reasoning Tasks, ICLR 2024
>
> [9] Wang et al., Benchmark Self-Evolving: A Multi-Agent Framework for Dynamic LLM Evaluation
>
> [10] Riemer et al., Learning to Learn without Forgetting by Maximizing Transfer and Minimizing Interference, ICLR 2019
>
> [11] Chaudhry et al., Efficient lifelong learning with A-gem. ICLR 2019

---

> > ### Comment · Reviewer_RbVT · 2024-11-23
> > **response to authors**
> >
> > Dear authors,
> >
> > Thank you for the very comprehensive responses!
> >
> > I am satisfied with the clarifications and updates for notation, metrics (and footnote), discussion of different pretrained backbones (thanks for pointing me to that section in the appendix and to the new results in the rebuttal), I think all this strengthens the narrative of the versatility of the proposed benchmark.
> >
> > re: proposed alternative for difficulty (W4), thanks for the response and additional experiment, I can clearly see through this discussion that my suggestion is correlated with the authors' approach. It would be great to add a sentence or footnote stating that these are heuristics that may capture data contamination to some extent, but do not faithfully measure contamination (they are just heuristics for difficulty, and difficulty can be due to different factors; data contamination being one of them). do you agree?
> >
> > re: the discussion on different levels of contamination and whether they affect the relative rankings or not, I agree with the sentiment that minor levels are ok (and in fact are present in realistic scenarios of interest), but I think it's very challenging to reason about the spectrum: what is the "degree of contamination" beyond which relative rankings are no longer meaningful. We can only speculate about this, since my understanding is that we don't have a ground-truth measurement of "degree of contamination", unless I missed some discussion here or in related work. It would be great to caveat such discussions where reflected in the paper by pointing out that this is an intuition (rather than a formal argument) if there is no concrete evidence.
> >
> > Thanks for all other discussions of related work and other issues and for already implementing the different levels of difficulty, these results are really nice. i've read through the entire responses and overall I believe that these strengthen the paper.
> >
> > Thanks again for your hard work!

---

> > > ### Author Response · Authors · 2024-11-24
> > > **Thank you for engaging in discussion with us; we made new updates to the latest manuscript.**
> > >
> > > Dear Reviewer RbVT,
> > >
> > > We are delighted to hear that you found our response satisfactory and our new results very nice!
> > >
> > > We would like to extend our heartfelt gratitude for your careful consideration and insightful feedback throughout the review process. Your comments have significantly contributed to enhancing the quality of our manuscript, and we truly value the time and effort you dedicated to this work.
> > >
> > > In response to the two points you raised, **we have revised the manuscript to include both discussions in Appendix E.1**, and we have referenced this in Line 181 of Section 3.1 in the main text. We hope that these updates address your concerns adequately.
> > >
> > > **If you find our revisions satisfactory**, we would greatly appreciate it if you could reconsider your score for our paper.
> > >
> > > Thank you once again for your invaluable feedback.
> > >
> > > Best regards,
> > >
> > > Authors

---

### Official Review · Reviewer_kAFn · 2024-11-09

**Soundness:** 2
**Presentation:** 2
**Contribution:** 2
**Rating:** 5
**Confidence:** 4

**Summary:**

This paper evaluates the existing CL approaches based on pre-trained foundation models over dynamic benchmarks, raising an important question of whether the existing methods benefit from data contamination, thereby resulting in higher overall continual learning performances, thus inducing less forgetting in the network over the sequential learning tasks.

**Strengths:**

1. The paper is well written and the problem setup is mostly clear.
2. This paper raises an interesting question involving the evaluation strategy of the recently proposed continual learning approaches exploiting pre-trained foundation models to mitigate catastrophic forgetting in the network while learning from sequentially arriving data.

**Weaknesses:**

While I find the paper interesting, questioning the existing evaluation strategy and raising a possibility that the recently proposed continual learning approaches involving pre-trained foundation models may benefit from data contamination, thereby suffering less from catastrophic forgetting over sequential learning, I have a few concerns about this paper.

1. This paper considers that the foundation models are pre-trained on large-scale datasets, like ImageNet-1k or ImageNet-21k, thereby, considers creating a dynamic benchmark with various datasets ranging across various types. However, the authors failed to consider that there are other large-scale datasets, such as JFT-300M, which might be the superset of the datasets considered by the authors in this paper, thereby raising a crucial question of what would happen in such a case. Furthermore, it is very much possible to train a foundation model with no supervision by scraping all images from the internet, thereby pre-training with a superset of these considered datasets, which raises the same question.

2. This paper does not consider online continual learning setup [1], which I believe is a crucial variant of continual learning research. However, I believe this paper cannot handle such a scenario.

[1] Prabhu, Ameya, Philip HS Torr, and Puneet K. Dokania. "Gdumb: A simple approach that questions our progress in continual learning." In Computer Vision–ECCV 2020: 16th European Conference, Glasgow, UK, August 23–28, 2020, Proceedings, Part II 16, pp. 524-540. Springer International Publishing, 2020.

**Questions:**

Please refer to weaknesses section.

---

> ### Author Response · Authors · 2024-11-22
> **Thank you for your review**
>
> We would first like to express our genuine appreciation for the time and effort the reviewer has dedicated to reviewing our manuscript. We believe these comments and questions provide valuable feedback and will greatly help us improve the overall quality of our work.
>
> Regarding the two weaknesses mentioned in the review, we respond to each in a separate comment section. Please let us know whether our response has sufficiently addressed your concerns. We are more than happy to engage in further discussion.

---

> ### Author Response · Authors · 2024-11-22
> **Response to W1**
>
> > [W1] This paper considers that the foundation models are pre-trained on large-scale datasets, like ImageNet-1k or ImageNet-21k, thereby, considers creating a dynamic benchmark with various datasets ranging across various types. However, the authors failed to consider that there are other large-scale datasets, such as JFT-300M, which might be the superset of the datasets considered by the authors in this paper, thereby raising a crucial question of what would happen in such a case. Furthermore, it is very much possible to train a foundation model with no supervision by scraping all images from the internet, thereby pre-training with a superset of these considered datasets, which raises the same question.
>
>
> We appreciate the reviewer's insightful comments regarding the contribution of our dynamic benchmark provided with more powerful foundation models pre-trained on larger datasets.
>
> To begin, we would like to humbly emphasize the core contribution of our CLDyB-pipeline -- its `dynamic` nature. The pipeline is exactly designed to  **re-generate task sequences from a expandable CLDyB-pool to challenge increasingly powerful CL learners and foundation models as the field advances**, while we initially plan to open-source CLDyB-seq as the fixed benchmark of task sequences that are generated to challenge the current state-of-the-art CL methods and foundation models.
>
> In addressing more powerful foundation models,
> - firstly, our CLDyB-pipeline **dynamically re-constructs** challenging task sequences via maximizing the reward function in Equation 4.
>     - We have indeed evaluated the CLDyB-pipeline using the **CLIP ViT-base** vision backbone, which was **pre-trained on a significantly larger, web-scraped dataset, YFCC100M**.
>     - The results, presented in **Appendix, Figure 11**, demonstrate that **the CLDyB-pipeline is still able to identify challenging CL sequences**, as evidenced by performance drops of -10%, -8%, and -2% in average final accuracy, average retention, and average learning accuracy, respectively,  of the CL methods.
> - secondly, our CLDyB-pipeline can also expand CLDyB-pool, introducing more diverse and challenging data over time.
>     - We have augmented the real-world CLDyB-pool with AI-generated data (see Lines 460-471 for details).
>     - The results, presented in Figure 6, show that **by expanding the CLDyB-pool as time evolves, our CLDyB-pipeline generates more challenging sequences that avoid performance saturation**.
>     - Additionally, during the response period, we conducted further experiments (see **Appendix Figure 14**) that confirm the augmented CLDyB-pool generates even more challenging sequences for the more powerful foundation model, CLIP-ViT-Base.
>
> In summary, the above experiments support the extendable success of our CLDyB-pipeline, fulfilling its design purpose of accommodating more powerful foundation models / CL learners. This is achieved through (a) dynamic task sequence re-searching (Equation (4)) and (b) continuous expansion of the CLDyB-pool.

---

> ### Author Response · Authors · 2024-11-22
> **Response to W2**
>
> >[w2] This paper does not consider online continual learning setup [1], which I believe is a crucial variant of continual learning research. However, I believe this paper cannot handle such a scenario.
>
> Thank you for raising this important question regarding compatibility with the online CL setup in [1]. Although we have focused primarily on the offline CL setting in the paper, we demonstrate below that **both the proposed CLDyB-pipeline and the discovered CLDyB-seq are readily applicable to the online setting in [1]**.
>
> - By principle, **benchmarks are independent of online/offline CL setups**.
> As defined in [1], the online setup imposes a constraint on the CL learner itself rather than the testing benchmark. Specifically, the **only difference** between the online [1] and offline CL setups **lies in whether the same data point can be used to optimize a CL model more than once (section 2.1, Online CL vs. Offline CL [1])**. This means that switching from offline to online affects how CL models are trained on each newly arrived task, but not what these tasks are or how they are constructed. **Therefore, by limiting the number of training epochs to 1 for CL models**:
>     -  **(a) The commonly challenging CLDyB-seq discovered in our paper can be directly used for evaluating CL models in an online setup**—just like how the same CIFAR-100 dataset can be used for evaluation in both online [1,2] and offline [3,4] CL setups.
>     - **(b) The CLDyB-pipeline is directly applicable to CL models in the online setup.** Our pipeline considers a CL model to challenge as a black-box function, denoted as $f^t =\mathcal{A}(f^{t−1} , \mathcal{T}^t)$, as detailed in Line 13 of Algorithm 3 (Appendix). Since we do not pose any explicit assumption about the internal training mechanism of the CL algorithm $\mathcal{A}$, adapting the CLDyB-pipeline from offline to online simply requires restricting the number of training epochs in the CL algorithm $\mathcal{A}$ to 1 for each incoming task. This restriction adheres to the online setup in [1], ensuring only a single pass over the data for optimization.
>
> - **By experiment**, we demonstrate the applicability of (a) Commonly Challenging (CC) CLDyB-seq established in the paper to evaluate GDUMB [1] and (b) our proposed CLDyB-pipeline to GDUMB [1] in search of an Individually Challenging (IC) CLDyB-seq.
>     - Compared with standard random task sequences (row 1), the noticeable performance drop, in AFA, ARA and ALA, indicates that both the commonly challenging CLDyB-seq and the CLDyB-pipeline are **indeed compatible with the online CL setup**. They are able to identify sufficiently challenging task sequences for the proposed online CL method, GDUMB, as outlined in [1].
>
>    - |  Settings | AFA | AR |ALA |
>      | -------- | -------- | -------- |-------- |
>      | online, standard random task sequences | 0.725     | -0.067     | 0.792      |
>      | online, CC CLDyB-seqs established in the paper    | 0.499     | -0.082     |0.581     |
>      | online, IC CLDyB-seqs found on GDUMB [1] only   | 0.437     | -0.089     |0.525     |

---

> > ### Comment · Reviewer_kAFn · 2024-11-26
> > **Misleading assumption about online learning**
> >
> > The authors assume that `online learning` means simply training the model with a single epoch. However, they overlook the fact that in online learning, the data distribution changes over time and is considered non-iid and is not known without observing the samples.
> >
> > The provided response, "Specifically, the only difference between the online [1] and offline CL setups lies in whether the same data point can be used to optimize a CL model more than once", is simply an oversimplification of the online learning setup. The proposed approach cannot handle online learning setup.

---

> ### Author Response · Authors · 2024-11-22
> **References**
>
> [1] Prabhu et al. GDumb: A Simple Approach that Questions Our Progress in Continual Learning. ECCV 2020.
>
> [2] Caccia et al. New Insights on Reducing Abrupt Representation Change in Online Continual Learning, NeurIPS 2019
>
> [3] Wang et al. Hierarchical decomposition of prompt-based continual learning: Rethinking obscured sub-optimality. NeurIPS, 2023
>
> [4] Zhang et al. Slca: Slow learner with classifier alignment for continual learning on a pre-trained model. ICCV 2023

---

> ### Author Response · Authors · 2024-11-25
> **We Look Forward to Hearing Your Feedback!**
>
> Dear Reviewer kAFn,
>
> We sincerely appreciate the time and effort you have dedicated to reviewing our work. Your insights and feedback are invaluable to us, and we are grateful for your contributions to the review process.
>
> As the discussion period is nearing its end, we wanted to gently follow up to see if our responses and revisions have addressed all your concerns.
>
> If you find our revisions satisfactory, we would be extremely grateful if you could reconsider your rating for our paper.
>
> Thank you once again!
>
> Best regards,
> Authors

---

> ### Comment · Reviewer_kAFn · 2024-11-26
> **Unclear Response to W1**
>
> > We have indeed evaluated the CLDyB-pipeline using the CLIP ViT-base vision backbone, which was pre-trained on a significantly larger, web-scraped dataset, YFCC100M.
>
> I believe that the authors are trying to simply overlook the previously raised concern: `Furthermore, it is very much possible to train a foundation model with no supervision by scraping all images from the internet, thereby pre-training with a superset of these considered datasets, which raises the same question.` The provided response does not fully address my concern.

---

> > ### Author Response · Authors · 2024-11-29
> > **The evolving nature of the world and the practical challenges of pre-training such a universally capable foundation model justifies the long-term significance of our dynamic benchmark**
> >
> > >Furthermore, it is very much possible to train a foundation model with no supervision by scraping all images from the internet, thereby pre-training with a superset of these considered datasets, which raises the same question.
> >
> > We sincerely thank the reviewer for your far-sighted feedback, which have greatly inspired us to refine our submission to further substantiate the long-term significance of our proposed pipeline in the future. We deeply appreciate how your feedback aligns with our vision of establishing a lasting impact through our dynamic benchmark.
> >
> > **The contribution of our proposed pipeline remains significant, as long as there exist data that CL algorithms starting from a pre-trained foundation model cannot handle well**—indicating room for improvement. By continuously adding such data to the CLDyB-pool and re-searching for new CLDyB-seqs, our benchmark evolves to challenge the latest advancements in both CL and foundation models. Below, we outline the **continued availability of such challenging data**.
> >
> > - Even if we could build an exceptionally strong foundation model at the current moment (e.g., by scraping all internet images):
> >    - Even the strongest foundation models, pre-trained on vast amounts of data, are still **susceptible to making non-trivial mistakes** [1,2].
> >    -  The fast-changing nature of the world and internet knowledge means today's capable foundation models likely struggle at **tomorrow's data**; this evolution combined with the difficulty of pre-training such a foundation model (which we will mention in the next) highlights two critical needs: (1) continual learning of such a foundation model, even powerful at this moment and  (2) a robust and adaptable evaluation framework, such as our CLDyB-pipeline, to foster ongoing advancements in CL algorithm development.
> >
> > - Moreover, it is practically non-trivial to build such a universally capable foundation model by scraping all internet images:
> >    - **Data storage and quality**: Storing and curating the vast amounts of internet data is not only computationally expensive but also fraught with issues like low-quality or noisy data[3].
> >    - **Model scaling challenges**: Scaling up model capacity to match the exponential growth of data poses significant engineering challenges[4].
> >    - **One-model-fits-all limitations**: Achieving a universal model capable of excelling across diverse tasks remains elusive, where domain-specific adaptations are always required following the current practice[2,5].
> >
> > **References**
> >
> > [1] Open AI, GPT-4 Technical Report, 2024
> >
> > [2] Radford et al., Learning Transferable Visual Models From Natural Language Supervision, ICML 2021
> >
> > [3] Longpre et al.,  A Pretrainer’s Guide to Training Data: Measuring the Effects of Data Age, Domain Coverage, Quality, & Toxicity, NAACL 2024
> >
> > [4] Meta, The Llama 3 Herd of Models, 2024
> >
> > [5] Abnar et al., Exploring the Limits of Large Scale Pre-training, ICLR 2022

---

> ### Author Response · Authors · 2024-11-29
> **The data in our CLyDB is non-i.i.d. & remains unknown until observed & is readily applicable to online CL (part 1/2)**
>
> We greatly appreciate the reviewer’s constructive feedback regarding the benchmark's compatibility with the online continual learning setup. Your comments are key to encouraging us to further explore the application of our proposed dynamic benchmark in a different CL scenario beyond offline class-incremental CL presented in our paper, thus broadening its applicability and potentially enhancing the impact of our work.
>
> Below, please kindly find our revised explanation of why our proposed benchmark also fits online continual learning. We hope this clarifies any remaining concerns and demonstrates our commitment to aligning our work with the reviewer’s thoughtful feedback. We are happy to engage further if additional clarification is needed.
>
> >... they overlook the fact that in online learning, the data distribution changes over time and is considered non-iid and is not known without observing the samples.
>
> Firstly, we totally agree with the reviewer and fully recognize that an important aspect of continual learning is the changing data distribution over time, where the data distribution is `non-IID` and  `remains unknown until they are observed`.
> - We apologize for not emphasizing this point in our previous response, as the above two criteria are fundamental prerequisites for any data considered in continual learning [2，3], whether in offline or online CL settings. **The data/task distribution in our constructed CLDyB-seq, designed for evaluating CL, strictly adheres to these two criteria**.
> - The data/task distribution in our constructed CLDyB-seq is `non-IID`, since
>   - the **sampling probability for tasks on the available classes varies between steps**:
>     - At each time step $t$, we adopt a greedy class sampling approach that **prioritizes sampling more difficult classes** as the K-way classification tasks, e.g., $p^t(\mathcal{T})\propto\prod_{q,p\in\{1,...,K\}}\Psi(\mathcal{C}_q,\mathcal{C}_p)$ (Equation 3), **instead of uniform sampling** for $K$ classes.
>     - More importantly,  **this 'difficultness' $\Psi$ is directly affected by the current CL model**'s feature space’s capacity $\mu^{t-1}$ through $\Psi(\mathcal{C}\_q,\mathcal{C}\_p)=\frac{1}{M}\sum\_{m=1}^{M}z(\text{cos}(\mu\_{m,p}^{t-1}, \mu_{m,q}^{t-1}))$,  **which evolves** over the time steps during CL (Lines 178-179).
>   -  At each $t$, **only previously unseen classes participate during task sampling**, as
>      - we strictly follow common settings in class-incremental continual learning where "all tasks have disjoint class label space" (Line 122).
>      - the **CLDyB-pool itself is expandable and time-evolving** in our dynamic pipeline, introducing more diverse and challenging data, hence non-iid-ness, in the underlying data distribution for constructing each task over the CL time steps. For more experimental details, please refer to Lines 460–471, Figure 6, and Figure 14 (Appendix).
> - The data/task distribution in our constructed CLDyB-seq `remains unknown to CL models until they are observed`.
>    - In **standard** CL training and evaluation protocols, **two steps** are repeated at each time step: (1) **a task $\mathcal{T}^t$ is constructed, (2)  training and evaluating the CL methods on that task**, $\mathbb{F}^{t}=\\{A\_m(f^{t-1}_m\,\mathcal{T}^t)\\}\_{m=1}^{M}$ (see Lines 120-122).
>    - Our dynamic benchmarking with the CLDyB-pipeline **adheres to this two-step procedure**. At each time step in CL, the CLDyB-pipeline (1) selects the next task $\mathcal{T}^t$ based on the latest states of the CL models, $\mathbb{F}^{t-1}$, (2) trains the CL models as $\mathbb{F}^{t}=\\{A_m(f^{t-1}\_m,\mathcal{T}^t)\\}\_{m=1}^{M}$, and evaluates on this newly selected $\mathcal{T}^t$ following standard protocols.
>    - As detailed above, the only difference occurs in how the next task is selected. We respectfully emphasize **the parallelism between the task selection process and CL training**. Consequently, the CL methods only observe the sequentially revealed tasks, **one by one over the time steps, with no access to past or future tasks and no prior knowledge about what the future tasks are**, consistent with standard CL settings (Line 123).

---

> ### Author Response · Authors · 2024-11-29
> **The data in our CLyDB is non-i.i.d. & remains unknown until observed & is readily applicable to online CL (part 2/2)**
>
> >The authors assume that online learning means simply training the model with a single epoch...
> >The provided response, "Specifically, the only difference between the online [1] and offline CL setups lies in whether the same data point can be used to optimize a CL model more than once **(section 2.1, Online CL vs. Offline CL [1])**", ...
>
> Given that both offline and online continual learning satisfy the above-mentioned criteria (in our part 1/2 response above), we would like to respectfully clarify our understanding of "online continual learning" as fully aligned with the literature (including [1] provided by the reviewer, [2,3] and others). Specifically,
>
> - As shown in Section 2.1 "Online CL vs. Offline CL" from [1], the difference between online and offline CL setups is the number of times with which each data point can be used to optimize the model.
> - Following these definitions from the referenced literature [1,2,3], we have empirically verified our approach's compatibility with online CL, for both CLDyB-seq and CLDyB-pipeline. Please see our detailed response in an earlier official comment here: https://openreview.net/forum?id=RnxwxGXxex&noteId=27MwlK6h0j.
>
>
>
> **References**
>
> [1] Prabhu et al. GDumb: A Simple Approach that Questions Our Progress in Continual Learning. ECCV 2020. **Section 2.1, Online CL vs. Offline CL**
> >Online CL v/s Offline CL: Note, the disjoint task formulation placed a
> restriction on the growing nature of the label space and inherently restricted the
> size of it, however, it did not put any constraints on the learner itself. Therefore,
> the learning paradigm may store task-specific samples coming from the stream
> depending on the space budget and then use them to update the parameters.
> Under this setting, in the online CL formulation, even though the learner is
> allowed to store samples as they come, they are not allowed to use a sample
> more than once for parameter update. Thus, the learner can not use the same
> sample (unless it is in the memory) multiple times at different iterations of the
> learning process. In contrast, offline CL allows unrestricted access to the entire
> dataset corresponding to a particular task (not to the previous ones) and one
> can use this dataset to learn the mapping by revisiting the samples again and
> again while performing multiple passes over the data
>
> [2] Zhou et al. Class-Incremental Learning: A Survey, TPAMI 2024. **Section 2.1 Problem Formulation, Online CIL**
> >Online CIL: Although data comes with stream format, the model can conduct multi-epoch training with each task, i.e., offline training within each task. There are some works addressing fully online (one-pass) CIL, where each batch can be processed once and then dropped. It is a specific case of the current setting, and we concentrate on the generalized CIL setting in this paper.
>
> [3] Masana et al. Class-incremental learning: survey and performance evaluation on image classification, TPAMI 2022. **2.2 General class-incremental learning setup**
> >Tasks consist of a number of classes, and learners are allowed to process the training data of the current task multiple times during the training session (also called the offline learning setting). We do not consider the online learning setting used in some papers in which each data sample is only seen once.

---

### Author Response · Authors · 2024-11-22
**Revision Summary**

We sincerely thank all the reviewers for their diligent efforts and high-quality reviews. If there are any additional questions or if further clarification is needed, please feel free to let us know. Your insights are highly valued.

We are delighted to note that the reviewers find that:

- Our method is novel `(Reviewer BrAw)`, well-motivated, and well-thought-out `(Reviewer RbVT)`, with mostly clear and easy-to-follow writing `(Reviewers kAFn, RbVT, BrAw)`.
- We contribute a new dynamic benchmark and a highly versatile task selection pipeline for continual learning methods with pre-trained models `(Reviewers kAFn, RbVT, BrAw)`, which lead to interesting results and are beneficial to later research `(Reviewers kAFn, RbVT)`.

In response to your valuable suggestions, we conducted additional experiments and revised our manuscript to include the following key new results:
- **Fig. 12** and **Fig. 13**, Appendix: We added experiments on using a mixture of pre-trained backbones with the CLDyB pipeline when searching for challenging sequences (suggested by `Reviewer RbVT`).
- **Fig. 14**, Appendix: We added experiments confirming that the augmented CLDyB pool can generate even more challenging sequences for the more powerful foundation model, CLIP-ViT-Base (suggested by `Reviewer kAFn`).
- **Appendix D.1**: We added experiments on evaluation with CLDyB-seq at various difficulty levels (suggested by `Reviewer RbVT`).

All changes made in the revised manuscript are highlighted in blue for your convenience.

---

### Meta-Review · Area_Chair_n87V · 2024-12-24

**Metareview:**

This paper points out the limitations of existing benchmarks used in continual learning (CL) and proposes a new framework called Continual Learning on Dynamic Benchmarks (CLDyB). The goal is to identify challenging tasks for CL methods and also determine, in a dynamic fashion, the order in which the tasks should be presented. The highly challenging task sequences provide more realistic evaluations of CL methods.

The reviewers largely appreciate the paper's contribution. They raised several questions too which the authors responded to in detail, and the reviewers seem to be mostly satisfied with the authors' response. There were some points of disagreement/misunderstanding with Reviewer kAFn regarding online CL and non-iid data. However, the authors' response clarified this satisfactorily.

The paper is a nice contribution in the area of CL, not by proposing a new method but in proposing robust evaluation strategies, which will be useful to this research community. In view of the reviews, the discussions, and my own reading of the paper, I think the paper surely meets the acceptance bar.

**Additional Comments On Reviewer Discussion:**

The reviewers were in general positive about the paper. Some concerns were expressed by Reviewer kAFn regarding online CL and non-iid data. However, the rebuttal provided a detailed clarification to these concerns. Although Reviewer kAFn didn't seem fully convinced by the rebuttal, having gone through the review and the rebuttal (and authors' follow-up comments/clarifications) myself, and having looked at the paper as well, I don't think the concerns of Reviewer kAFn to be critical but rather a misunderstanding.

The decision was made after factoring in all these points.

---

### Decision · Program_Chairs · 2025-01-22

Accept (Poster)